# PLUMAGE: Probabilistic Low-rank Unbiased Min-Variance Gradient Estimator Framework for Efficient Large Model Training

## Abstract

Accelerator memory and network constraints are dominant bottlenecks when training large language models (LLMs) with billions of parameters. Recently, Low-rank gradient estimators have been successfully applied by methods such as Galore and Flora for LLM training on consumer hardware, by compressing gradients and optimizer tensors. However, the underlying gradient estimation methods are biased or subject to a high variance. Moreover, low-rank optimizer states, such as the first and second moments under the previous subspace, become misaligned whenever the projection is updated. This misalignment can lead to instabilities during training with low-rank gradients. We propose PLUMAGE: Probabilistic Low-rank Unbiased Minimum-vAriance Gradient Estimator. PLUMAGE can be applied as a drop-in replacement for low-rank LLM training without introducing new hyperparameters beyond the chosen rank $r$ and the update interval $\tau$. In addition, we resolve the misalignment of low-rank statistics. We apply PLUMAGE as a drop-in replacement for the fixed top-$k$ components estimate used in Galore to observe how the gradient bias-variance trade-off impacts the optimization of LLMs. The resulting PLUMAGE+Adam shrinks the full-rank optimization's gap in the pre-training evaluation loss by 33% on average across models, with a 34% improvement on the commonsense benchmark for the 1B models. In finetuning tasks, the average training loss gap across the GLUE benchmark is shrunk by 28% — without retuning the full-rank learning rate and within a similar computational and memory footprint as Galore. Alternatively, in our 1B pretraining benchmark PLUMAGE+Adam surpassed the terminal loss of Galore within 30% fewer steps.

## 1 Introduction

Natural language modeling has benefited from training large language models (LLMs), which consist of billions of parameters on massive amounts of unsupervised data. This requires distributed training algorithms employing thousands of interconnected high-end accelerators to speed up the training time. Moreover, some of the largest state-of-the-art models do not fit in a single device's memory. Meanwhile, the historical progress of scaling the capabilities of high-bandwidth memory and device connectivity lags behind progress in scaling the compute capabilities of modern accelerators (Gholami et al., 2024). Thus, significant engineering and research efforts have been made to improve large model training efficiency. For example, distributed computing (Shoeybi et al., 2019; Rajbhandari et al., 2020; Zhao et al., 2023), structured sparsity (Hubara et al., 2021; Chmiel et al., 2022), and applications of low-precision numerical formats Courbariaux et al. (2016); Banner et al. (2018); Chmiel et al. (2020); Dettmers et al. (2021); Blumenfeld et al. (2024); Chmiel et al. (2025).

As models grow in size, the memory and connectivity limitations of modern accelerators emerge as a critical bottleneck, impeding the ability of the research community to train and fine-tune state-of-the-art models without relying on access to expensive hardware and infrastructure. Addressing these constraints is essential for reducing entry barriers for future research. One recent avenue of work aims to enable the training and fine-tuning of large models with limited accelerator memory overhead. A common strategy is Parameter-Efficient Fine-Tuning (PEFT), where trainable adapters are inserted into frozen large models to learn a new task. For example, Hu et al. (2021) proposed attaching low-rank parametrized adapters to the linear layers of a given model. This approach lowers

the optimizer's state memory requirements compared to full model training, while reducing the risk of overfitting on small datasets. This strategy can also incorporate memory and computation gains using low-bit numerical formats (e.g., Dettmers et al. (2023)). However, PEFT methods rely on having a pre-trained model and are less suited when one wishes to train the entire model.

Recently, Zhao et al. (2024) (GALORE) proposed projecting gradients to a low-rank subspace. The method projects the dense linear layers' weight gradients, reducing the optimizer's state memory – without enforcing a low-rank weight structure. This method was also shown to yield a potentially better compression rate and lower terminal loss compared to other PEFT methods. More importantly, the authors showed how LLMs can be trained from scratch on widely available consumer-grade hardware with limited memory. Notably, the projection at the core of the method can be viewed as a low-rank gradient estimation based on the top-$k$ singular vectors, periodically computed via Singular Value Decomposition (SVD, Eckart & Young (1936)) of the linear weight gradient.

These results led us to question whether there is a better low-rank gradient estimation alternative for training neural networks, given that the gradients are not necessarily of low rank. Moreover, fixing the projection to the span of the top-$k$ singular vectors can lead to a significant accumulation of bias during the optimization process, which is known to be detrimental (Gupta et al., 2015; Chmiel et al., 2020; 2025). Our contributions can be summarized as follows:

- We derive a novel $k$-sparse probabilistic low-rank unbiased minimum-variance gradient estimator (PLUMAGE). Our approach relies on an efficient and fixed-rank sampling strategy without replacement. In addition, it requires permanent storage of only a one-sided projection matrix per weight, similar to the top-k gradient estimator used in Zhao et al. (2024).
- We develop an alignment method for the first and second moments used by stateful optimizers such as ADAM. Our alignment strategy mitigates the adverse effects of periodic projection subspace updates during training, enhancing training stability with low-rank gradient estimators.
- We demonstrate a significant improvement in convergence rate, terminal validation loss, and in the post-training evaluation metrics, when using our low-rank gradient estimator as a drop-in replacement for the top-k estimator — without tuning the full-rank learning rate or adding hyperparameters other than the rank of choice and projection update interval.

## 2 RELATED WORK

This section reviews the foundational and closely related work necessary to contextualize our approach. We will also explain why it is important to distinguish between memory-efficient gradient estimation and memory-efficient optimization.

**Momentum and Adaptive Optimizers**. Given the learning rate $\eta$ and the gradient $\mathbf{G}_t = \nabla \mathcal{L}_t$ in step $t$, the vanilla stochastic gradient descent (SGD) updates the weight matrix in each layer according to

$$\mathbf{W}_{t+1} = \mathbf{W}_t - \eta \mathbf{G}_t. \tag{1}$$

Adding a momentum term with coefficient $\beta_1$, to dampen stochastic gradient noise (SGDM, Rumelhart et al. (1986)) yields the following.

$$\mathbf{M}_t = \beta_1 \mathbf{M}_{t-1} + (1 - \beta_1)\mathbf{G}_t \quad \text{and} \quad \mathbf{W}_{t+1} = \mathbf{W}_t - \eta \mathbf{M}_t. \tag{2}$$

Adaptive methods precondition the update by incorporating second-order information. For example, ADAM uses a diagonal preconditioner (i.e., a per-coordinate scale), by tracking the second moment estimate Kingma & Ba (2017),

$$\mathbf{V}_t = \beta_2 \mathbf{V}_{t-1} + (1 - \beta_2)\mathbf{G}_t^{\circ 2}, \tag{3}$$

where $\square^{\circ 2}$ denotes an element-wise square. ADAM's update rule (bias correction factors omitted for brevity) is

$$\mathbf{W}_{t+1} = \mathbf{W}_t - \frac{\eta \mathbf{M}_t}{\sqrt{\mathbf{V}_t} + \epsilon}. \tag{4}$$

**Memory-Efficient Optimizers**. Despite its popularity and advantages, ADAM incurs a high computational and memory overhead. State compression is an alternative, straightforward approach to

reduce the optimizer's state without changing the optimizer (e.g., quantization Dettmers et al. (2021)). Other preconditioning strategies can lead to lower state memory. For example, Adafactor tracks a factorized estimate of the second moment with sublinear memory in the size of the matrix parameters Shazeer & Stern (2018). Other optimizers do not use or store a second-moment/preconditioner. For example, Chen et al. (2023) used updates that are computed as the sign of the current gradient and the momentum. The sign operation normalizes the coordinate-wise magnitude. It was shown that the extended Lion-$\mathcal{K}$ family solves a constrained optimization of the loss under $l_\infty$. This family includes the recent promising MUON optimizer (Jordan et al., 2024; Chen et al., 2025; Liu et al., 2025) that employs semi-orthogonal updates by applying Newton-Shults iterations before applying weight updates. In general, such methods are orthogonal to our low-rank gradient approximation.

**Sparse Gradients**. Sparse gradient methods focus on updating only a subset of model parameters, reducing compute, communication, and memory costs (e.g., Aji & Heafield (2017); Chmiel et al. (2022)). Recently, Muhamed et al. (2024) proposed GRASS that leverages structured gradient sparsity to reduce its optimizer state memory and computational cost. GRASS is constructed as an unbiased and minimum variance row-sparse gradient estimator by sampling rows in proportion to their norms. However, its optimal variance is established under a multinomial sampling *with-replacement*. This allows high norm rows to be sampled multiple times and thus leads to a higher variance vs. non-replacement sampling. Moreover, the authors suggest that computing the optimal sampling probabilities without replacement is not tractable. In addition, a row-sparse approach leads to instabilities during training, prompting the usage of heuristics such as momentum restarts and periodic learning-rate warm-ups when the sampled row indices are updated.

**Low-Rank Gradients for Communication Compression.** In distributed settings, low-rank gradients are applied to compress the data transfer during gradient synchronization between the distributed collective workers, before each optimization step. For example, Wang et al. (2018) (ATOMO) proposed a min-variance unbiased low-rank gradient estimator that reduces communication by sparsifying the gradient's singular values and transmitting only the surviving SVD components. ATOMO's sampling strategy produces a rank-$k$ gradient *on average* and requires communication of *both* the left and right projections. Later, PowerSGD (Vogels et al., 2020) leveraged the power-iteration method to maintain an approximation of the top-$k$ gradient singular vectors, reducing the cost of applying SVD on the gradients before each communication, while adding a feedback mechanism to incorporate the previous step error in the next gradient communication.

**Low-Rank Gradients for Memory-Constrained Optimization.** GALORE and FLORA exploit the low-rank gradient structure to reduce device memory costs (Zhao et al., 2024; Hao et al., 2024) when training and finetuning large models that do not fit in the device memory. Recall that GALORE, as previously mentioned, uses a biased fixed top-$k$ singular-vector projected gradient estimator. In contrast, FLORA utilized reconstructible random Gaussian projections (via random generator seed reuse). FLORA authors also use ADAFACTOR instead of ADAM to further reduce the low-rank optimizer state memory footprint. While FLORA's gradient estimation is unbiased, the random projection matrices yield an additive variance proportional to the number of trainable parameters. Thus, it is less suited for training large models from scratch. The authors address this caveat by setting a low update frequency, trading off bias for variance while focusing on small model training and fine-tuning tasks. Recently, Shamshoum et al. (2024) (DROPACT) proposed applying a random projection (similarly to FLORA) to the linear layer's inputs during the forward phase before storing them for the backward phase to further reduce input memory for the weight gradient. This approach reduces activation, gradients, and optimizer state memory as the gradients are only projected back for weight update. We will later show (Section 4.3) that FLORA typically leads to a higher loss than both GALORE and our method. Moreover, attempts to find a comparable regime to pretrain large models with FLORA+ADAM were unsuccessful. We attribute this primarily to the high variance incurred by sampling high-dimensional random projections.

**Full-Rank Training with Low-Rank States**. A recent line compresses the optimizer states while utilizing a *full-rank gradient* for the weight update. FIRA (Chen et al., 2024) stores low-rank moments in a projector-defined subspace (e.g., top singular subspace as in GALORE), but applies the gradient residual so that updates remain full-rank. APOLLO (Zhu et al., 2024) computes channel-wise (or tensor-wise) adaptive factors from random projected moments and applies them to the full-rank gradient. Since both methods require instantiating a full-rank gradient, they *cannot* reduce gradient memory or communication volume in distributed training. Thus, they are closer to memory-efficient optimizers and are orthogonal to our proposed gradient estimation framework.

In contrast, a low-rank gradient estimator reduces the dimensionality of the gradient signal and is optimizer-agnostic. This approach results in simultaneous savings in gradient-related memory while enabling a lower communication overhead, which is critical for training large models on lower tiers of accelerator hardware.

# 3 PLUMAGE: A PROBABILISTIC LOW-RANK UNBIASED MINIMUM vARIANCE GRADIENT ESTIMATOR

In this section, we develop our gradient estimator framework following similar Minimum Variance Unbiased Estimator (MVUE) considerations as in Alain et al. (2015); Wang et al. (2018); Chmiel et al. (2022). In addition, we require a fixed-rank estimator constraint, leading to a deterministic computational and memory cost that is essential for practical applications. Moreover, we seek a compact one-sided projection matrix to reduce the required storage and match GALORE. Finally, we establish PLUMAGE's efficient sampling strategy and develop the tools for deploying PLUMAGE in stateful optimizers such as ADAM.

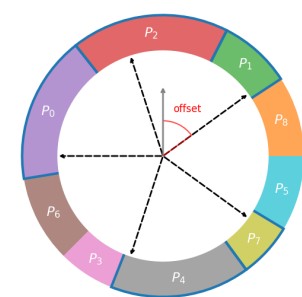

Figure 1: "Wheel-of-Fortune" sampling with $k = 5$ arms

## 3.1 LOW-RANK MIN-VARIANCE UNBIASED ESTIMATOR

Given the matrix $\mathbf{G} \in \mathbb{R}^{m \times n}$ and its singular values decomposition $\mathbf{G} = \sum_{i=1}^{n} \sigma_i \mathbf{u}_i \mathbf{v}_i^\top$, where $n \leq m$ and $\boldsymbol{\sigma} = \{\sigma_i\}_{i=1}^{n}$ is ordered from largest to smallest singular values. The classic low-rank estimator based on the top-$k$ singular vectors with $k \leq n$ is given by $\hat{\mathbf{G}}_{\text{top}-k} = \sum_{i=1}^{k} \sigma_i \mathbf{u}_i \mathbf{v}_i^\top$ and is known to have the minimal mean square error Eckart & Young (1936). However, the truncation of the tail components results in gradient bias, which is detrimental to the optimization process, as errors are accumulated over multiple training iterations (see Appendix D). Therefore, we consider a low-rank gradient estimator with the general form

$$\hat{\mathbf{G}} = \sum_{i=1}^{n} \frac{1}{p_i} \mathcal{I}_i \sigma_i \mathbf{u}_i \mathbf{v}_i^\top, \tag{5}$$

where $\mathcal{I}_i$ denotes a random 0/1 variable which determines the inclusion of the $i^{\text{th}}$ component, while $p_i$ is a scalar constant. We aim to find a distribution for $\mathcal{I}_i$ such that the following properties hold:

1) Unbiased gradient estimator:

$$\mathbb{E}\hat{\mathbf{G}} = \mathbf{G} \Leftrightarrow p_i = \mathbb{E}\mathcal{I}_i \tag{6}$$

2) Deterministic rank $k$:

$$\sum_{i=1}^{n} \mathcal{I}_i = k \Rightarrow \sum_{i=1}^{n} p_i = k \tag{7}$$

3) Minimal variance:

$$\min_{\hat{\mathbf{G}}} \mathbb{E}\left[\text{Var}\left(\hat{\mathbf{G}}\right)\right] \tag{8}$$

We solve the constrained optimization to minimize the variance with the following results (a complete derivation is given in Appendix A).

The variance for a low-rank estimator from Eq. (5) is

$$\mathbb{E}\left[\text{Var}\left(\hat{\mathbf{G}}\right)\right] = \sum_{i=1}^{n}\left[\left(\frac{1}{p_i} - 1\right)\right]\sigma_i^2. \tag{9}$$

Next, to obtain the optimal sampling probabilities, one must first compute the fixed-rank budget,

$$r^*\left(k, \boldsymbol{\sigma}\right) = \underset{r \in \{0,..,n\}}{\arg\min} r \text{ s.t. } \frac{(k-r)\sigma_{r+1}}{\sum_{i=r+1}^{n} \sigma_i} < 1. \tag{10}$$

Then, the minimum-variance unbiased sampling probabilities are given by

$$p_i = \begin{cases} 1 & \text{, if } i \leq r^* \\ \frac{(k-r^*)\sigma_i}{\sum_{j=r^*+1}^{n} \sigma_j} & \text{, if } i > r^* \end{cases}. \tag{11}$$

We observe that $\mathcal{I}_i$ need not be independent. Thus, we can employ any sampling strategy that satisfies the $k$-sparse condition in Eq. (7) under Eq. (11). Formally, let $k \leq n$ and $\{p_i\}_{i=1}^{n}$ a series of inclusion probabilities, with $p_i \in (0, 1]$ and $\sum_{i=1}^{n} p_i = k$. We wish to sample without replacement exactly $k$ distinct indices $I \triangleq \{i_1 \ldots i_k\}$ such that $\forall i \in \{1 \ldots n\} : \mathbb{P}(i \in I) = p_i$.

We employ a "wheel-of-fortune" sampling trick with $k$ equidistant arms (Fabian, 2024) to efficiently sample and construct the projection matrix in linear time complexity over the number of singular values. In essence, we first shuffle the order of $p_i$, then each $p_i$ is represented by a sector that is proportional to $\frac{p_i}{\sum_{k=1}^{n} p_k}$. Finally, the arms are randomly rotated with a uniformly distributed shift, selecting all the indices in a single step. The sampling algorithm is illustrated in Fig. 1. Computing $p$ and sampling can be implemented efficiently in $\mathcal{O}(\min(m, n))$ and is negligible compared to SVD computation (see Algorithms 2 and 3 in the appendix).

## 3.2 ONE-SIDED PLUMAGE

Next, we construct an estimator using only a one-sided projection and show that our one-sided estimator maintains the MVUE properties. Without loss of generality, we assume our gradient estimator is based on a left-sided projection $\mathbf{P}$. The projection matrix is constructed by stacking the sampled singular vectors,

$$\mathbf{P} = \left[ \underbrace{\mathbf{u}_1 \mathcal{I}_1}_{\text{column 1}} \quad \cdots \quad \underbrace{\mathbf{u}_n \mathcal{I}_n}_{\text{column } n} \right]. \tag{12}$$

Similarly, right-sided projections stack $\mathbf{v}_i$ singular vectors as rows. The left-side projection of matrix $\mathbf{G}$ is given by

$$\boldsymbol{P}^\top \boldsymbol{G} = \sum_{k=1}^{n} \sigma_k \boldsymbol{P}^\top \mathbf{u}_k \mathbf{v}_k^\top = \sum_{k=1}^{n} \sigma_k \mathcal{I}_k \boldsymbol{\delta}_k \mathbf{v}_k^\top, \tag{13}$$

where $\boldsymbol{\delta}_k \in \{0, 1\}^n$, $\boldsymbol{\delta}_k[i] = \begin{cases} 1 & i = k \\ 0 & \text{else} \end{cases}$.

Given the diagonal scaling matrix $\mathbf{D} = \text{Diag}(p_1, p_2, ..., p_d)$, the left sided PLUMAGE estimator is

$$\hat{\boldsymbol{G}} = \boldsymbol{P}\boldsymbol{D}^{-1}\boldsymbol{P}^\top \boldsymbol{G} = \sum_{k=1}^{n} \sigma_k \mathcal{I}_k \boldsymbol{P}\boldsymbol{D}^{-1}\boldsymbol{\delta}_k \mathbf{v}_k^\top = \sum_{i=1}^{n} \frac{1}{p_k} \mathcal{I}_k \sigma_k \boldsymbol{u}_k \mathbf{v}_k^\top. \tag{14}$$

Since Eq. (14) is equivalent to the original formulation from Eq. (5), the one-sided projection has the same properties as the two-sided variant.

## 3.3 INCORPORATING PLUMAGE IN OPTIMIZERS

We apply PLUMAGE to compress the optimizer's first and second momentum, which are common statistics tracked by optimizers such as ADAM and SGDM.

### 3.3.1 LOW-RANK MOMENTS AND ADAM WEIGHT UPDATE

The low-rank moments are given by

$$\mathbf{M}_t^{\lfloor t} = \beta_1 \mathbf{M}_{t-1}^{\lfloor t-1} + (1-\beta_1) \mathbf{P}_t^\top \mathbf{G}_t \quad \text{and} \quad \mathbf{V}_t^{\lfloor t} = \beta_2 \mathbf{V}_{t-1}^{\lfloor t-1} + (1-\beta_2) \left(\mathbf{P}_t^\top \mathbf{G}_t\right)^{\circ 2}, \tag{15}$$

where $\mathbf{P}_t \in \mathbb{R}^{m \times r}$ denotes the left projection matrix composed from the singular vectors of $\mathbf{G}_t$, and $\square^{\lfloor t}$ denotes the low rank moment from time step $t$. The simplified low-rank ADAM update rule is

$$\mathbf{W}_{t+1} = \mathbf{W}_t - \eta \mathbf{P}_t \mathbf{D}_t^{-1} \frac{\mathbf{M}_t^{\lfloor}}{\sqrt{\mathbf{V}_t^{\lfloor} + \epsilon}}, \tag{16}$$

where $\mathbf{D}_t$ is the diagonal matrix containing its sampled singular values, as explained in Section 3.2.

### 3.3.2 PERIODIC PROJECTION UPDATE

We aim to amortize costly SVD operations by reusing the same projection across multiple optimizer steps, similarly to Zhao et al. (2024). Given the hyperparameter $\tau \in \mathbb{N}^+$, the interval length between subsequent SVD. For convenience, we define the index mapping $k_t = \lfloor t/\tau \rfloor$. We define the optimizer statistics as

$$\mathbf{M}_t^{\lfloor k_t} = \beta_1 \mathbf{M}_t^{\lfloor k_t} + (1 - \beta_1) \mathbf{P}_{k_t} \mathbf{G}_t \quad \text{and} \quad \mathbf{V}_t^{\lfloor k_t} = \beta_2 \mathbf{V}_t^{\lfloor k_t} + (1 - \beta_2) (\mathbf{P}_{k_t} \mathbf{G}_t)^{\circ 2}. \tag{17}$$

The projection $\mathbf{P}_{k_t}$ and scaling factors $\mathbf{D}_{k_t}$ are sampled once after computing SVD over $\mathbf{G}_{k_t}$ according to the probability $\boldsymbol{p}_{k_t}$ computed according to Eq. (11). Moreover, PLUMAGE needs only to store the diagonal elements in $\mathbf{D}_{k_t}$ up to the chosen rank, contributing to minor memory overhead compared to the baseline GALORE memory cost. Note that in this case, $\boldsymbol{p}_{k_t}$ is no longer proportional to $\mathbf{G}_{k_t}$'s singular values. Thus, we lose the minimum variance promise of PLUMAGE, yet the estimate remains unbiased. However, if the gradient subspace changes slowly during optimization, we can still benefit from variance reduction.

One can also define a resampling interval $\kappa \in \mathbb{N}^+$ hyperparameter to refresh the projection and scaling coefficients without recomputing the SVD (i.e., $\kappa \leq \tau$). This can help balance subspace exploration and SVD compute overhead under the uncertainty of how well the sampled subspace retains the information from the observed gradients. In our experiments, sampling projection once throughout the SVD update interval works well. Yet, we found that overly frequent projection updates may hurt the utility of the moment statistics in the optimization process. We leave the exploration of this tradeoff for future work and set $\kappa = \tau$ in our experiments. In addition, we explored methods for measuring and utilizing the subspace correlation with the recent gradients in Appendix B.

### 3.3.3 STATISTICS REALIGNMENT OF THE FIRST AND SECOND MOMENTS

Stateful optimizers such as ADAM, when using low-rank estimators such as GALORE, suffer from a crucial subspace alignment issue when the gradient's projection is updated during training. In essence, the low-rank statistics represent moment estimates tracked in different subspaces. Formally, let $\mathbf{G}^{\lfloor t} = \mathbf{P}_t^\top \mathbf{G}$ then, we aim for $\mathbf{M}^{\lfloor t} = \mathbb{E}\left[\mathbf{G}^{\lfloor t}\right]$ and $\mathbf{V}^{\lfloor t} = \mathbb{E}\left[\left(\mathbf{G}^{\lfloor t}\right)^{\circ 2}\right]$. However, given two arbitrary projection metrices $\mathbf{P}_1 \in \mathbb{R}^{m \times r_1}$ and $\mathbf{P}_2 \in \mathbb{R}^{m \times r_2}$, it is clear that $\mathbf{M}^{\lfloor 1} \neq \mathbf{M}^{\lfloor 2}$ and $\mathbf{V}^{\lfloor 1} \neq \mathbf{V}^{\lfloor 2}$ For instance, if $\mathbf{P}_1 \perp \mathbf{P}_2$, then $\mathbf{M}^{\lfloor 1}$ has no information on $\mathbf{M}^{\lfloor 2}$. Moreover, misalignment can occur even if the subsequent projections span the same subspace (e.g., a rotation of the singular vectors). Therefore, updating the projections without statistics realignment may lead to spurious weight updates in the following weight updates (until the new data is sufficiently represented in the first and second moment estimators). Similarly to Hao et al. (2024), one can initialize $\mathbf{M}^{\lfloor 2}$, after updating the projection, by projecting the previous estimate onto the shared subspace between the two projections to realign the first-moment statistics,

$$\mathbf{M}^{\lfloor 2} \approx \mathbf{P}_2^\top \mathbf{P}_1 \mathbf{M}^{\lfloor 1} \tag{18}$$

Notably, unlike Hao et al. (2024), under PLUMAGE's $\mathbf{P}_i$ are orthonormal matrices. Thus, $\mathbf{B} = \mathbf{P}_2^\top \mathbf{P}_1$ projects $\mathbf{M}^{\lfloor 1}$ onto the intersection of subspaces induced by the projections $\mathbf{P}_i$. The resulting $\mathbf{M}^{\lfloor 2}$ magnitude will be proportional to the cosine of the principal angle between the two subspaces since $\|\mathbf{B}\| \leq 1$. Moreover, when $\mathbf{P}_i$ spans the same subspace, the transformation rotates the moment without distortion since $\|\mathbf{B}\| = 1$. For $\mathbf{V}^{\lfloor 2}$, one cannot simply initialize $\mathbf{V}^{\lfloor 2} = \mathbf{B}\mathbf{V}^{\lfloor 1}$ after the change in projections, since it will produce negative entries. Since finding the optimal inverse projection is not trivial, we approximate $\mathbf{V}^{\lfloor 2}$ as an initial guess.

$$\mathbf{V}_{ij}^{\lfloor 2} = \mathbb{E}\left[\mathbf{B}\mathbf{G}^{\lfloor 1}\right]_{ij}^2 = \sum_{k,l} B_{ik} B_{il} \mathbb{E}\left[G_{kj}^{\lfloor 1} G_{lj}^{\lfloor 1}\right] \approx \sum_{l,k} B_{ik} B_{il} \mathbb{E}\left[\left(G_{kj}^{\lfloor 1}\right)^2\right] = \left[\mathbf{B}^{\circ 2}\mathbf{V}^{\lfloor 1}\right]_{ij}, \tag{19}$$

where in the $\approx$ step we approximated the gradient's second-moment matrix to be diagonal, i.e., $\mathbb{E}\left[G_{kj}G_{lj}\right] \approx 0$ for $k \neq l$. Such a diagonal approximation is commonly used in both theory and practice (Kingma & Ba, 2017; ichi Amari et al., 2018). In Section 4.1, we empirically demonstrate that our initialization produces better results than simply assuming $\mathbf{M}^{\lfloor 1} \approx \mathbf{M}^{\lfloor 2}$ and $\mathbf{V}^{\lfloor 1} \approx \mathbf{V}^{\lfloor 2}$ when updating the gradient projection. The full definition of PLUMAGE+ADAM with momentum realignment is given in Algorithm 5 of the appendix. In the experiment section below, we refer to this combination as PLUMAGE for brevity.

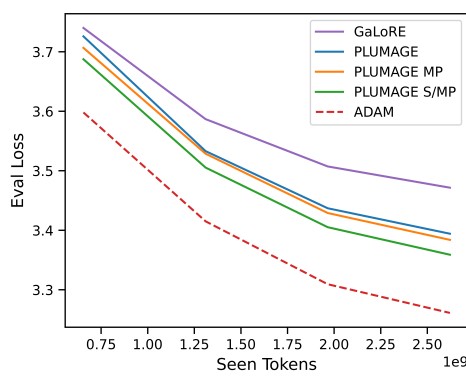
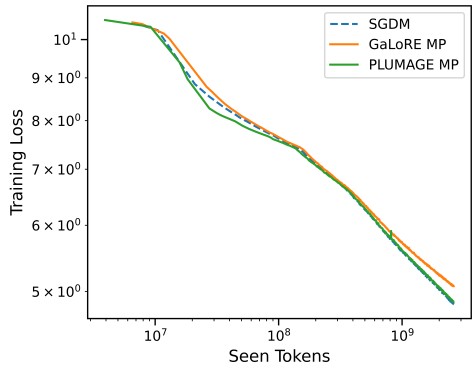

(a) Llama 130M Ablation: perplexity vs. tokens.  (b) Llama 350M C4 loss, w/o second moment.

Figure 2: Qualitative evaluation of PLUMAGE

## 4 EXPERIMENTS

### 4.1 PLUMAGE ABLATION STUDY

As in Zhao et al. (2024), we evaluate the optimization performance of our PLUMAGE+ADAM optimizer by pre-training LLaMA with varying sizes on the C4 English dataset from scratch. In all C4 English experiments, we rely on the processed version of the "Colossal Cleaned Crawl", which is readily available in the Huggingface datasets library (Wolf et al., 2019; for AI, 2020; Raffel et al., 2019). As in Zhao et al. (2024), we use a sequence length of 256 with the T5 tokenizer from Raffel et al. (2019); however, shorter sequences are stringed together to yield 256 tokens instead of padding.

Table 1: Llama 130M Ablation: terminal loss and accuracy after 2.6B tokens.

|  | Loss | | Acc |
|---|---|---|---|
|  | Train | Eval |  |
| ADAM | 3.256 | 3.256 | 38.7 |
| GALORE | 3.473 | 3.471 | 36.5 |
| PLUMAGE | 3.402 | 3.400 | 37.1 |
| PLUMAGE$_{MP}$ | 3.386 | 3.385 | 37.3 |
| PLUMAGE$_{S/MP}$ | 3.378 | 3.377 | 37.4 |

The ablation results are presented in Table 1 with a token vs. loss curve in Fig. 2a. We include the base PLUMAGE (Eq. (17)), add the projection realignment methods: The 'first **M**oment **P**rojection (Eq. (18), $\Box_{MP}$), and the '**S**econd moment realignment' (Eq. (19), $\Box_{S/MP}$). An optimization gap between the low-rank gradient optimizers and the full-rank ADAM is still apparent. However, PLUMAGE$_{S/MP}$ shrinks the gap between the GALORE and Adam baseline by a factor of $\sim 2$ when the hyperparameters are set according to the full-rank ADAM training regime (that Zhao et al. (2024) found to be optimal). Additional ablations over the rank and projection update interval and other qualitative experiments can be found in Appendix C.

### 4.2 PLUMAGE DEBIAS IMPACT

While PLUMAGE is unbiased, applying it to ADAM does not produce unbiased gradients due to the use of the non-linear adaptive scaling factor based on the **V** statistic. The **V** term contributes to gradient bias with the ADAM update as well, thus degradation observed in Table 1 is expected. To better understand how bias impacts optimization with low-rank gradients, we train a medium-sized Llama variant with 350M parameters on 2.6B tokens from the C4 English dataset as before. However, since the adaptive step size is biased, we use stochastic gradient descent with momentum (SGDM). This is achieved by disabling the second-moment term in Adam and comparing it with the adapted versions of PLUMAGE$_{MP}$ and GALORE$_{MP}$ variants. We use momentum $\beta = 0.9$ for all optimizers and a learning rate of $0.1$. In addition, we fix the projection intervals to 200 steps and the rank to 128, while the hidden dimension of the model is 1024. In Section 4.2, we observe the training loss on a log scale. Our version of SGDM+PLUMAGE$_{MP}$ convergence rate is on par with the full-rank SGDM optimizer, while it is clear that SGDM+GALORE$_{MP}$ bias is impeding the optimization. For a self-contained read, we include a simple quadratic scaler biased optimization example from Chmiel et al. (2025) in Appendix D.

### 4.3 FINE-TUNING WITH LOW-RANK GRADIENTS

#### 4.3.1 GLUE BENCHMARK

Following Zhao et al. (2024) and Hao et al. (2024), we evaluate low-rank optimization on GLUE natural understanding benchmark (Wang et al., 2019). We fine-tune all parametrized layers in the Roberta base model (Liu et al., 2019) except for its embedding layers, while low-rank gradient estimators are used for all linear layers' weights. We fix the rank $r = 8$, and reuse the hyperparameters as suggested in Zhao et al. (2024). In particular, the update interval $\tau = 100$ and the recommended $\alpha$ scales for GALORE (recall that PLUMAGE does not use $\alpha$). We optimize the model for 30 epochs to ensure the loss has converged. Learning rates are tuned per task for each method to yield the best possible results, as done in Zhao et al. (2024). This allows us to compare the terminal loss and to evaluate the estimators' impact on the optimization gap. The results are reported in Table 2, where we present the best metrics for accuracy and loss for each method in every task. It is also notable that using PLUMAGE leads to lower or equivalent terminal loss and next token prediction accuracy compared to both FLORA and GALORE.

Table 2: Fine-tuning RoBERTa-Base on GLUE benchmark: best terminal accuracy (loss).

| | MNLI | QQP | QNLI | SST-2 | CoLA | STS-B | MRPC | RTE | Mean |
|---|---|---|---|---|---|---|---|---|---|
| ADAM | 87.60 (4e-3) | 89.35 (2e-3) | 92.86 (2e-3) | 94.27 (4e-3) | 63.58 (4e-3) | 90.92 (0.03) | 92.93 (1e-3) | 75.81 (3e-3) | 85.91 (4e-3) |
| FLORA | 87.07 (0.25) | 87.77 (0.15) | 92.33 (0.15) | 94.84 (0.07) | 59.31 (0.13) | 89.95 (0.22) | 91.42 (0.16) | 71.84 (0.24) | 84.32 (0.16) |
| GALORE | 85.78 (0.29) | 85.46 (0.21) | 91.95 (0.11) | 92.78 (0.08) | 62.32 (0.07) | 90.75 (0.09) | 91.36 (0.02) | 78.70 (0.04) | 84.89 (0.11) |
| PLUMAGE$_{S/MP}$ | 87.58 (0.22) | 88.23 (0.13) | 92.62 (0.06) | 94.72 (0.05) | 61.57 (0.03) | 90.94 (0.09) | 91.29 (5e-3) | 78.34 (0.02) | 85.66 (0.08) |

#### 4.3.2 FINE-TUNING LARGE LANGUAGE MODELS

To demonstrate the impact of low-rank gradient estimators on LLM finetuning. We employ supervised fine-tuning on Llama 3.1-8B (Grattafiori et al., 2024) on Tulu-3 SFT mixture following the regime from Lambert et al. (2025). Specifically, we used a sequence max length of 4192 and a 128 global batch size. The maximal learning rate was $5 \cdot 10^{-6}$ with 3% warmup steps budget followed by a linear decay schedule. In addition, we use the open-sourced SFT script from Huggingface with a standard ChatML template

Table 3: Training statistics for Llama 3.1-8B SFT on Tulu3

| | Acc | Loss | Ppl |
|---|---|---|---|
| Baseline[1] | 70.7 | 1.31 | 3.71 |
| ADAM[2] | 79.2 | 0.77 | 2.16 |
| GALORE | 74.2 | 1.05 | 2.86 |
| PLUMAGE$_{S/MP}$ | 75.2 | 0.97 | 2.64 |

(Werra et al., 2020). The final training statistics are reported in Table 3 with a significant advantage for PLUMAGE in both training loss/perplexity and next-token accuracy. In Table 5, we also evaluate each model on popular commonsense and reasoning benchmarks (Storks et al., 2020; Hendrycks et al., 2021; Jin et al., 2020; Cobbe et al., 2021) using EleutherAI LM Evaluation Harness (Gao et al., 2024). Results show a small downstream accuracy improvement over GALORE in most tasks, possibly due to the relatively small number of weight update steps and the small learning rate used in the SFT regime, which may limit the impact of bias in gradients on the downstream accuracy.

### 4.4 PRE-TRAINING LANGUAGE MODELS FROM SCRATCH

The pre-training process of LLMs involves digesting trillions of tokens, setting a high bar for model quality. It is challenging to replicate such experiments, particularly on consumer-grade equipment. However, exploring the dynamics of training large models from scratch is crucial for facilitating research into more efficient methods for training at scale. Thus, we attempt to train the models to a point where we can gain insight into the performance of different low-rank gradient optimizers.

Table 4: Validation perplexity for C4 pre-training.

| | 130M | 350M | 1B |
|---|---|---|---|
| ADAM | 25.95 | 19.02 | 14.3 |
| GALORE | 30.18 | 24.08 | 17.03 |
| PLUMAGE$_{S/MP}$ | 28.73 | 21.81 | 16.29 |
| Rank/Hidden | 256/768 | 256/1024 | 512/2048 |
| Tokens Seen | 2.6B | 7.9B | 26.2B |
| Optimizer Steps | 20K | 60K | 100K |
| Warmup Steps | 2K | 6K | 10K |

We follow Zhao et al. (2024) and pre-train several Llama variants with 130M, 350M, and 1B parameters on the C4 English dataset as in Section 4.1. We reuse the training regime and the best hyperparameters for GALORE and ADAM. To confirm the choice of the learning rate for PLUMAGE, we run a learning rate grid search for the 130M and 350M models and find that $10^{-3}$ (same as ADAM)

Table 5: Commonsense: Llama 3.1-8B before and after Tulu-3 SFT

| | BoolQ | PIQA | SciQ | HellaS | WinoG | ARC-E | ARC-C | OBQA | MMLU | MedQA | GSM8K | Mean |
|---|---|---|---|---|---|---|---|---|---|---|---|---|
| Baseline[1] | 60.43 | 50.54 | 22.4 | 26.3 | 46.33 | 26.01 | 26.62 | 27.4 | 23.88 | 27.34 | 50.34 | 35.24 |
| ADAM[2] | 82.54 | 82.10 | 95.30 | 80.48 | 74.11 | 83.12 | 55.80 | 46.80 | 63.77 | 56.95 | 72.10 | 72.10 |
| GALORE | 83.46 | 81.39 | 94.80 | 79.76 | 74.35 | 82.07 | 53.92 | 44.80 | 64.12 | 59.94 | 56.10 | 70.43 |
| PLUMAGE$_{S/MP}$ | 83.00 | 81.50 | 95.00 | 79.90 | 73.80 | 82.20 | 54.01 | 45.80 | 64.09 | 59.94 | 58.15 | 70.67 |

Table 6: Commonsense[3]: Llama 350M and 1B pretrained variants (C4 dataset)

| | Size | BoolQ | PIQA | SciQ | HellaS | WinoG | ARC-E | ARC-C | OBQA | MMLU | MedQA | Mean |
|---|---|---|---|---|---|---|---|---|---|---|---|---|
| ADAM | | 58.99 | 68.44 | 64.80 | 39.77 | 51.93 | 43.64 | 26.11 | 28.40 | 22.94 | 27.73 | 43.28 |
| GALORE | 350M | 60.15 | 64.64 | 59.80 | 33.19 | 50.20 | 39.56 | 22.27 | 29.00 | 23.02 | 27.65 | 40.95 |
| PLUMAGE$_{S/MP}$ | | 55.93 | 66.49 | 62.10 | 35.22 | 50.75 | 40.61 | 24.49 | 27.80 | 22.95 | 27.65 | 41.4 |
| ADAM | | 57.58 | 73.01 | 71.10 | 54.63 | 56.27 | 51.52 | 26.88 | 33.00 | 24.58 | 28.67 | 47.72 |
| GALORE | 1B | 46.79 | 68.93 | 65.9 | 45.12 | 52.41 | 44.32 | 25.34 | 29.4 | 23.18 | 28.12 | 42.95 |
| PLUMAGE$_{S/MP}$ | | 56.48 | 70.46 | 66.90 | 47.09 | 51.62 | 47.35 | 25.26 | 29.60 | 23.17 | 27.97 | 44.59 |

consistently produces close to the best result. Both GALORE and PLUMAGE use the same fixed projection update interval of $\tau = 200$ steps and a fixed rank for each model as in Zhao et al. (2024). The results in Table 4 show that PLUMAGE consistently outperforms GALORE — without having to sweep over additional hyperparameters and without impacting the total train time. Moreover, training PLUMAGE was stable over multiple seeds, while in our experiments, GALORE suffers from sharp fluctuations in the training loss, with the loss diverging in some seeds. The training curves can be found in Fig. 8 of the Appendix. Notably, while PLUMAGE has a similar run time, it surpassed GALORE's loss within 50% and 70% of the training steps budget for 350M and 1B models. In Table 6, we evaluate the pretrained models on the commonsense and reasoning benchmark suite as in Section 4.3.2 with a maximal sequence length of 256 tokens. These results suggest that Plumage gains may become significant with model size and more optimization steps.

## 5 DISCUSSION

**Summary.** We presented PLUMAGE under a low-rank gradient estimation framework. Our method samples single-sided projections without replacement and yields a $k$-sparse and unbiased minimum variance estimator of the gradient. We also showed how to improve the accumulated projected statistics used by optimizers during training, when the projection is updated. We demonstrate PLUMAGE superiority in memory-efficient optimization against other low-rank estimators, such as the top-k estimator used by GALORE with similar computational and memory costs. Notably, PLUMAGE can improve storage and communication in a wider context. The following applications follow a similar optimization process as in our experimental setup, yet we leave their in-depth exploration to future work.

**Gradient and activation memory compression.** Similar to Shamshoum et al. (2024), applying strictly right-sided PLUMAGE projections to linear layers' inputs during the forward phase can reduce the storage of activation, gradient, and optimizer state. This is mathematically equivalent to Eq. (17), albeit the projection cost is added to each forward pass instead of just once per optimizer step (while the periodic gradient for SVD requires instantiating a full-rank gradient).

**Communication reduction.** The gradient's size can become a bottleneck in common LLM training scenarios. For example, in a fully-sharded distributed setting, gradient buffers are often distributed across the collective's devices' memory, resulting in a significant communication overhead (Rajbhandari et al., 2020; Zhao et al., 2023). Even with mixed precision training, gradient accumulation buffers are often kept in full-precision to avoid truncation errors. In such cases, low-rank gradients reduce communication and memory overheads. Notably, PLUMAGE can also recover the full-rank estimate in a data-parallel setting (i.e., via independently sampled projection matrix per-worker). This is appealing when the communication bottleneck is significant (e.g, federated learning McMahan et al. (2023)). Moreover, PLUMAGE fixed-rank budget is also crucial for practical applications sensitive to compute and memory variance.

**LLM usage disclosure.** LLMs were primarily used as a writing aid, polishing specific segments, providing notation suggestions, initial reference lookup, etc.

---

[1] pretrained only, no SFT

[2] using SFT reference checkpoint

[3] GSM8K is omitted from Table 6 since 256 sequence length is too short for the standard 8-shot test

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

# TECHNICAL APPENDICES AND SUPPLEMENTARY MATERIAL

## A  PROBABILISTIC LOW-RANK UNBIASED MIN VARIANCE DERIVIATION

We wish to minimize the expected variance given by,

$$\min_{\hat{\mathbf{G}}} \mathbb{E}\left[\mathrm{Var}\left(\hat{\mathbf{G}}\right)\right] = \min_{\mathbf{p}=\{p_1,\ldots,p_n\}} \mathbb{E}\mathrm{Tr}\left[\left(\hat{\mathbf{G}} - \mathbb{E}\hat{\mathbf{G}}\right)^{\top}\left(\hat{\mathbf{G}} - \mathbb{E}\hat{\mathbf{G}}\right)\right]. \tag{20}$$

We plug the estimator definition from Eq. (5) into Eq. (20) and use the orthogonality of singular vectors to obtain $\mathbf{u}_i^{\top}\mathbf{u}_j = \mathbf{v}_i\mathbf{v}_j^{\top} = \delta_{ij}$,

$$\mathbb{E}\mathrm{Tr}\left[\left(\sum_{i=1}^{n}\left(\frac{1}{p_i}\mathcal{I}_i - 1\right)\sigma_i\mathbf{u}_i\mathbf{v}_i^{\top}\right)^{\top}\left(\sum_{j=1}^{n}\left(\frac{1}{p_j}I_j - 1\right)\sigma_j\mathbf{u}_j\mathbf{v}_j^{\top}\right)\right] \tag{21}$$

$$\mathbb{E}\left[\sum_{i,j=1}^{n}\left(\frac{\mathcal{I}_i}{p_i} - 1\right)\left(\frac{\mathcal{I}_j}{p_j} - 1\right)\mathrm{Tr}\left(\mathbf{v}_i\mathbf{u}_i^{\top}\mathbf{u}_j\mathbf{v}_j^{\top}\sigma_i\sigma_j\right)\right] = \mathbb{E}\left[\left(\sum_{i=1}^{n}\left(\frac{\mathcal{I}_i}{p_i} - 1\right)\sigma_i^2\right)\right] \tag{22}$$

$$= \sum_{i=1}^{d}\mathbb{E}\left[\left(\frac{\mathcal{I}_i}{p_i} - 1\right)^2\right]\sigma_i^2 = \sum_{i=1}^{d}\left[\left(\frac{1}{p_i} - 1\right)\right]\sigma_i^2 \tag{23}$$

Taking into account the sparsity condition in Eq. (7), we define $\mathbf{p}$ as the indicator probabilities vector that minimizes the variance in Eq. (23). We find $\mathbf{p}$ by solving the following problem

$$\min_{\mathbf{p}} \sum_{i=1}^{n}\left[\left(\frac{1}{p_i} - 1\right)\right]\sigma_i^2 \text{ s.t. } \sum_{i=1}^{n}p_i = k, p_i \in [0,1] \tag{24}$$

which is equivalent to

$$\min_{\mathbf{p}} \sum_{i=1}^{n}\frac{\sigma_i^2}{p_i} \text{ s.t. } \sum_{i=1}^{n}p_i = k, p_i \leq 1 \tag{25}$$

Note that we cannot allow for $p_i = 0$ since then the optimization objective is undefined. To solve Eq. (25), we write the Lagrangian

$$\sum_{i=1}^{n}\frac{\sigma_i^2}{p_i} + \mu\sum_{i=1}^{n}p_i + \sum_{i=1}^{n}\lambda_i p_i \tag{26}$$

where $\mu > 0, \lambda_i > 0$ are the Lagrangian factors, and after differentiating by $p_i$ we get

$$0 = -\frac{\sigma_i^2}{p_i^2} + \mu + \lambda_i \Rightarrow p_i = \frac{\sigma_i}{\sqrt{\mu + \lambda_i}} \tag{27}$$

Note that $\lambda_i > 0$ only if we hit the inequality constraints, i.e., $p_i = 1$. To satisfy the constraints, we first solve

$$r^*\left(k, \boldsymbol{\sigma}\right) = \arg\min_{r \in \{0,..,d\}} r \text{ s.t. } \frac{(k-r)\sigma_{r+1}}{\sum_{i=r+1}^{n}\sigma_i} < 1 \tag{28}$$

Then, we return the following solution

$$p_i = \begin{cases} 1 & , \text{if } i \leq r^* \\ \frac{(k-r^*)\sigma_i}{\sum_{j=r^*+1}^{n}\sigma_j} & , \text{if } i > r^* \end{cases}. \tag{29}$$

## B  MEASURING PROJECTION FITNESS VIA PRINCIPAL ANGLES AND ADAPTIVE PROJECTION INTERVALS

Since applying SVD to each gradient at every step is costly, similarly to GALORE, we amortize the cost of SVD over multiple optimization steps, reusing the projection from an old gradient and

updating it within some interval. However, since projected state optimizers (Zhao et al., 2024; Liang et al., 2024) converge despite using uncorrected statistics with alternating projections, we hypothesize that the dominant directions are relatively stable at least in some layers. Zhao et al. (2024) also suggested that some layers require less frequent updates than others and proposed a per-layer adaptive interval controller to reduce the computational overhead of SVD. Specifically, the controller computes the cosine similarity between the first singular vectors of two subsequent projections to determine if the interval can be extended based on a fixed threshold. We argue that the suggested metric can be sensitive and unreliable. For example, if the ordering of singular vectors changes while the subspace remains unchanged, the proposed metric will return 0. In addition, when the correlation between subsequent subspaces breaks, the metric cannot be used to shrink the interval to reflect uncertainty.

We devise an alternative approach based on the framework of principal angles Zhu & Knyazev (2013). Namely, we calculate the cosine of the principal angle between the spanning subspaces by computing the singular values of $\mathbf{P}_1^\top \mathbf{P}_2 \in \mathbb{R}^{r_1 \times r_2}$ as follows,

$$\boldsymbol{\sigma} \leftarrow \mathrm{SVD}\left(\mathbf{P}_1^\top \mathbf{P}_2\right). \tag{30}$$

Finally, we take the mean cosine angle to represent the intersection between the two induced subspaces.

$$\rho = \mathrm{Mean}(\boldsymbol{\sigma}) \tag{31}$$

This approach offers a principled and robust measure for the overlap between subsequent projections during training. This approach allows our controller to manage the period between projection updates reliably. One specific advantage is the ability to reduce the interval when the assumptions regarding subsequent overlap break during training. In practice, we define a simple hysteresis controller with 3 threshold hyperparameters, $\gamma_{\mathrm{shrink}}, \gamma_{\mathrm{expand}}, \gamma_{\mathrm{reset}}$ that can be set by observing how well the training loss with a low-rank gradient estimator follows the loss of a standard optimizer and the recoded values of the principle angles with some fixed interval.

Table 7: Token-level validation perplexity for pre-training with low-rank gradient estimators.

| Method | 130M | 350M | 1B |
|---|---|---|---|
| ADAM | 25.95 | 19.02 | 14.3 |
| GALORE | 30.18 | 24.08 | 17.03 |
| PLUMAGE$_{\mathrm{S/MP}}$ | 28.73 | 21.81 | 16.29 |
| PLUMAGE$_{\mathrm{S/MP/A}}$ | 29.26 | 21.79 | 16.24 |
| Rank/Hidden | 256/768 | 256/1024 | 512/2048 |
| Tokens Seen | 2.6B | 7.9B | 26.2B |
| Optimizer Steps | 20K | 60K | 100K |
| Warmup Steps | 2K | 6K | 10K |

The adaptive controller is given in Algorithm 1. In practice, the overlap can be approximated well with only the top 64 singular vectors from the old sampled projection matrix and the top 64 singular vectors from the freshly computed singular vectors of the new gradient (without sampling) to reduce the computational overhead of SVD when extracting the mean cosine principal angles. In addition, we find that allowing the interval to grow unconstrained undermines the original purpose of accelerating training, as it leads to degradation in the loss convergence rate and terminal value. Thus, we set a $\tau_{max} = 5\%$ of the total steps while $\tau_{\min} = \tau_{\mathrm{initial}} = 200$. Finally, we set $\gamma_{\mathrm{shrink}}, \gamma_{\mathrm{expand}}, \gamma_{\mathrm{reset}} \leftarrow 0.4, 0.6, 0.3$ by observing the statistics during training, as can be seen in Figs. 5 and 6. We report the results in Table 7. Using our conservative adaptive interval hyperparameters, the equal-weighted average interval length over all layers in the models grew from the initial interval of 200 steps to $\sim 425$ steps and $\sim 1000$ steps for the 1B and 350M models. We observed minor terminal loss improvements in 1B and 350M models, potentially due to improved moment estimation with longer intervals. Ultimately, this avenue requires significant manual tuning to produce real train time gains while avoiding degradation in loss. Thus, we leave further exploration of this topic for future study utilizing larger models where the SVD overhead dominates the computation time.

## C  ADDITIONAL QUALITATIVE EXPERIMENTS

### C.1  RANK AND INTERVAL ABLATION STUDIES

We perform additional ablation experiments on the rank and SVD interval impact on the Llama 130M variant. The results are presented in Fig. 3. Similar to Zhao et al. (2024), we find that setting the gap interval too short or too wide leads to degradation in the optimization process. The short intervals potentially lead to poor first and second-moment estimates due to frequent projection updates. In

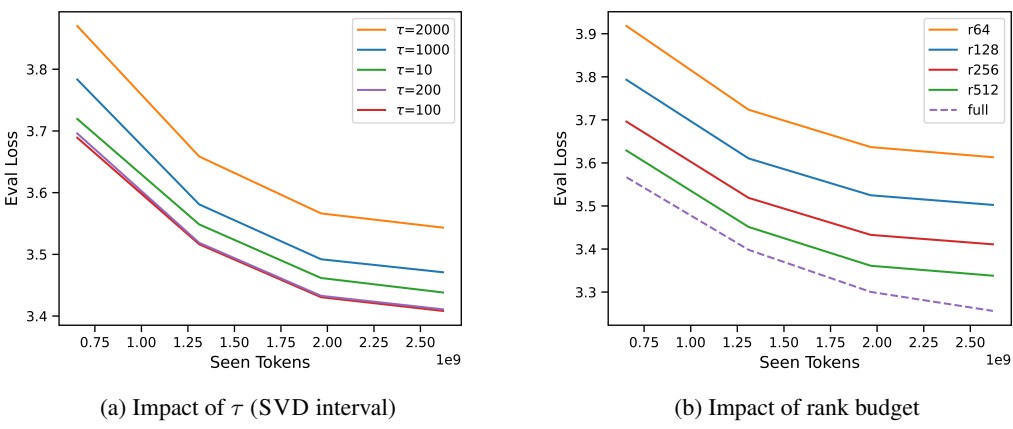

(a) Impact of $\tau$ (SVD interval)

(b) Impact of rank budget

Figure 3: Ablation studies comparing validation loss Llama 130m on C4

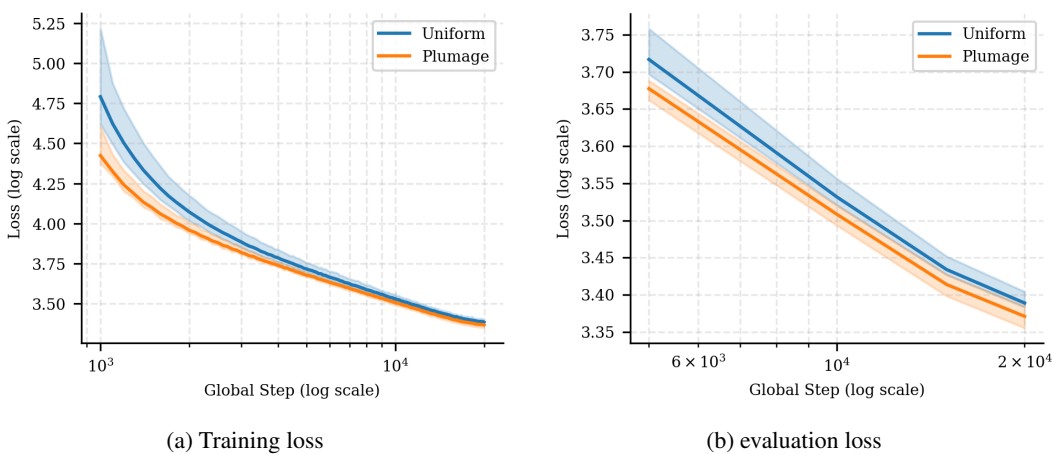

(a) Training loss

(b) evaluation loss

Figure 4: Ablation studies comparing Uniform and Plumage sampling loss of Llama 130m on C4 over 5 different seeds

---

**Algorithm 1** Adaptive projection interval

---

**inputs:** subsequent projections $\mathbf{P}_{t-1}, \mathbf{P}_t \in \mathbb{R}^{m \times r}$, interval configurations $\tau_{\min}, \tau_{\max}, \tau_{\text{initial}}$, and thresholds $\gamma_{\text{reset}} \leq \gamma_{\text{shrink}} < \gamma_{\text{expand}} \in [0, 1]$
**initialize:** $\tau_0 \leftarrow \tau_{\text{initial}}, t \leftarrow 0$
$\rho_t \leftarrow \text{mean\_cosine\_principle\_angle}(\mathbf{P}_{t-1}, \mathbf{P}_t)$          $\triangleright$ Eqs. (30) and (31)
**if** $\rho_t < \gamma_{\text{reset}}$ **then**
    $\tau_t \leftarrow \tau_{\min}$
**else if** $\rho_t < \gamma_{\text{shrink}}$ **then**
    $\tau_t \leftarrow \max(\tau_{t-1}/2, \tau_{\min})$
**else if** $\rho_t > \gamma_{\text{expand}}$ **then**
    $\tau_t \leftarrow \min(2 \cdot \tau_{t-1}, \tau_{\max})$
**else**
    $\tau_t \leftarrow \tau_{t-1}$
**end if**
$t \leftarrow t + 1$
**return:** $\tau_t$

---

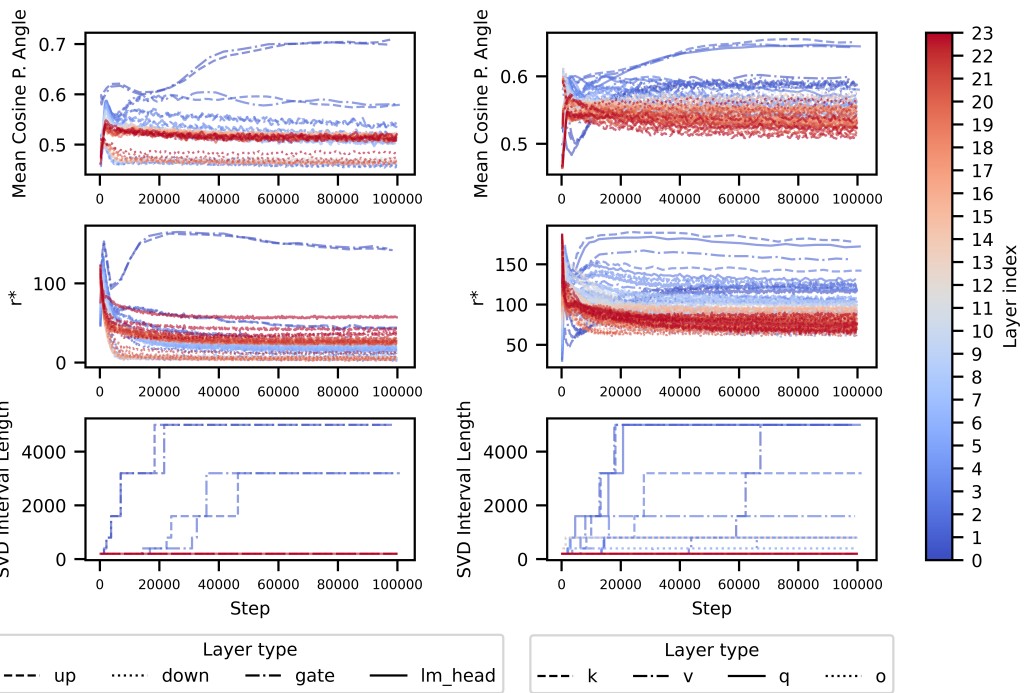

Figure 5: Tracking $r*$ (deterministic rank) and $\rho$ (mean cosine principal angle) during pretraining of Llama 1B on C4 with PLUMAGE$_{\text{S/MP/A}}$

addition, since $\mathbf{B} = \mathbf{P}_2^\top \mathbf{P}_1 \leq 1$ (see Section 3.3.3), the effective learning rate is also reduced. Finally, the choice of rank should be as large as one can fit onto the device memory, as can be seen in Fig. 3b. In Fig. 4 we show how sampling accoding to Plumage yields lower variance and better loss compared to using a uniform sampling distribution over the projection components.

## C.2 LOW-RANK COVERAGE QUALITY

During training, we monitor the development of $r*$ and $\rho$ for different layers. In Fig. 5, we show the progress of the metrics over time, and in (Figs. 6 and 7), we show the average values per-layer throughout the entire training. By tracking $r*$ in proportion to the total rank. We find that some layers, such as the attention projection layers, are more suitable for rank reduction. In contrast, the attention MLP down-projection layer parameters are the least amenable. This invites further exploration of rank budget allocation between different layers. For instance, stacking the $QKV$ linear layers into a

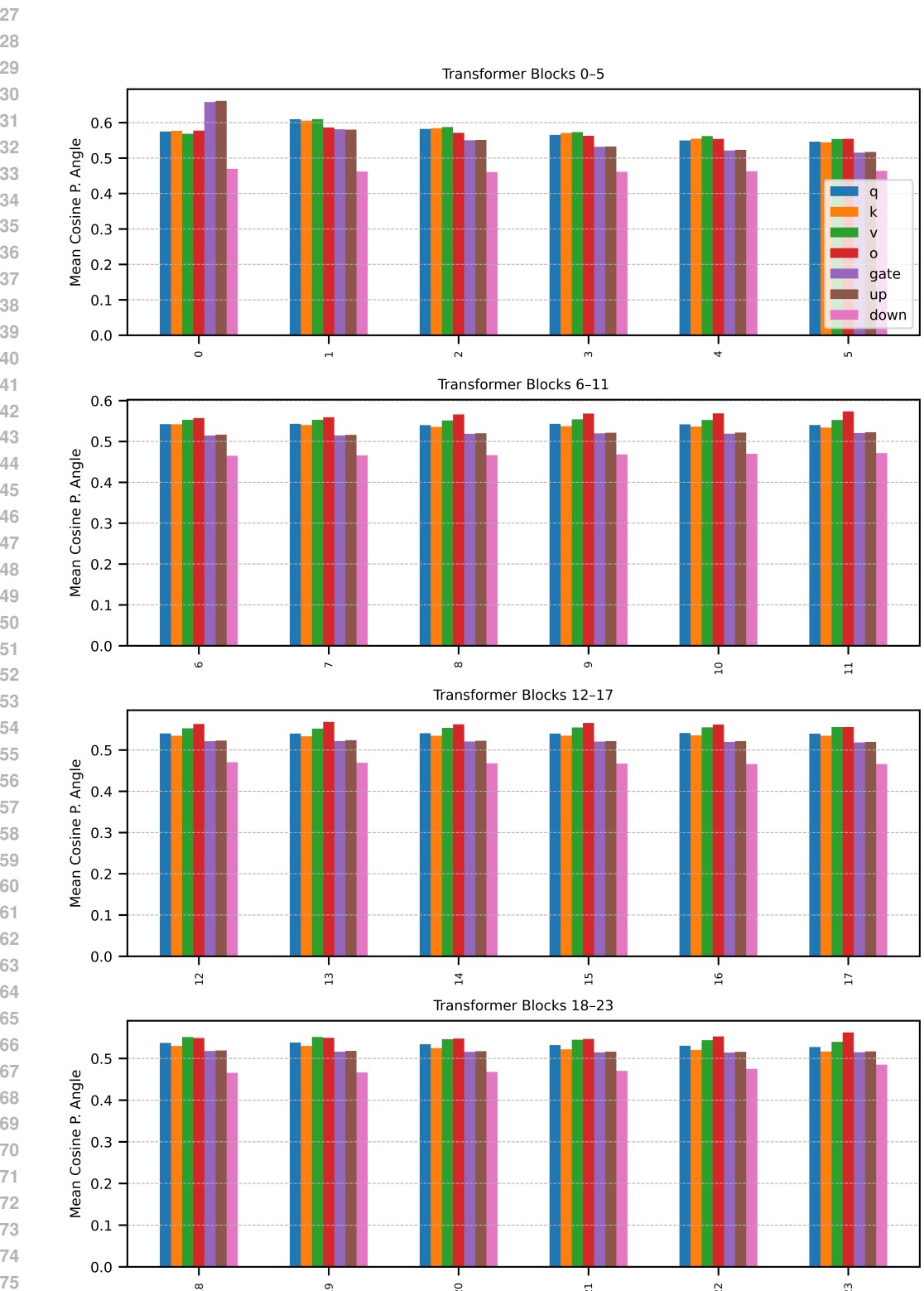

Figure 6: Mean $\rho$ observed during LLama2-1B pretraining on C4 with rank=512

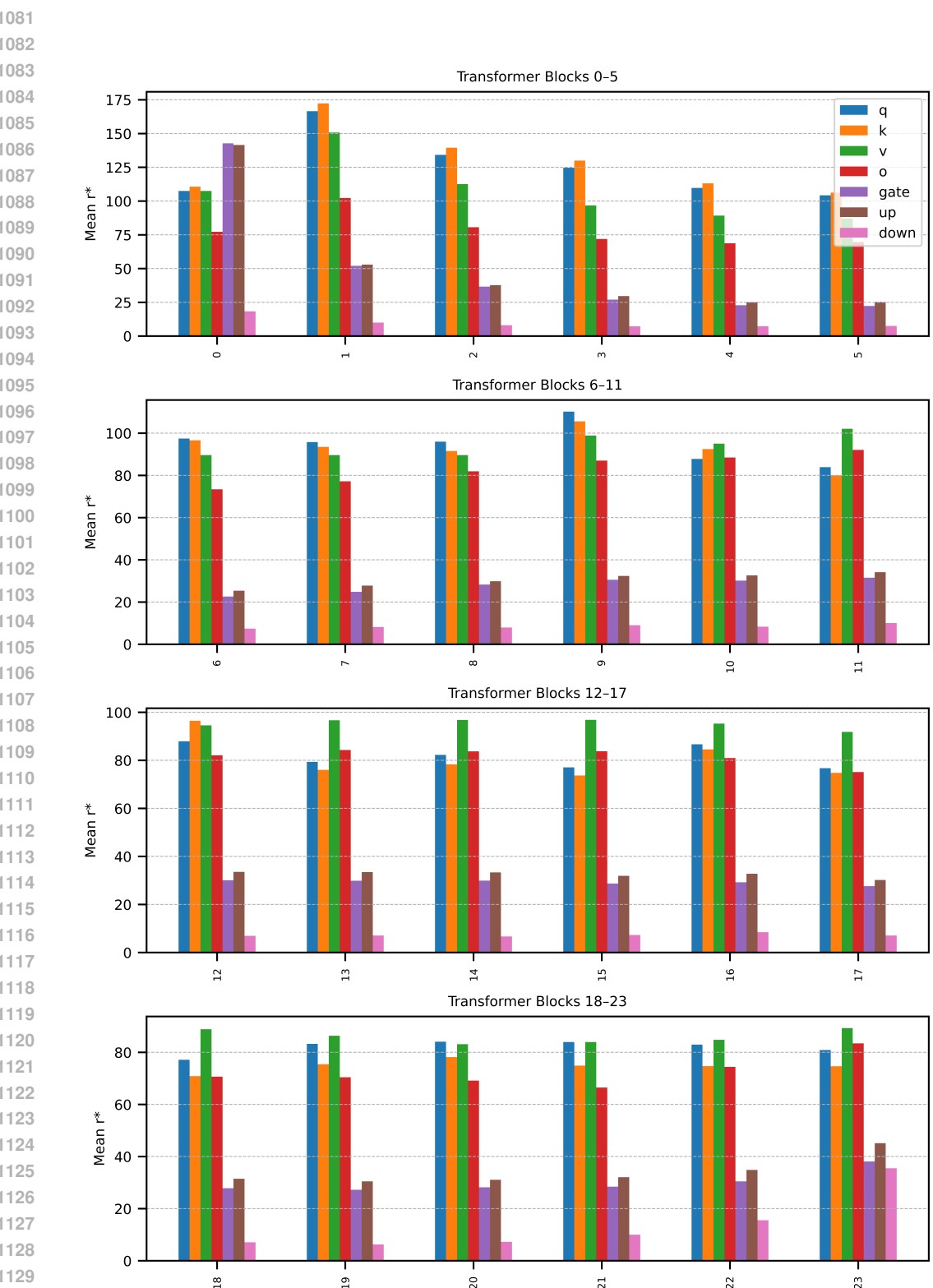

Figure 7: Mean $r^*$ observed during LLama2-1B pretraining on C4 with rank=512

single matrix allows us to share a single low-rank projection matrix without impacting convergence while reducing SVD computational overhead. Moreover, the memory savings can be translated to increasing the rank of the less amenable layers, such as the MLP down layer, to improve the convergence within the same memory budget. We leave this topic to be explored in future work.

# D GRADIENT BIAS ACCUMULATION

We include this section from Chmiel et al. (2025) for completeness to exemplify the problem with biased quantization schemes, in a simple scalar optimization problem with a quadratic loss

$$L(\theta) = \tfrac{1}{2} \lambda (\theta - \theta^*)^2, \implies \nabla L(\theta) = \lambda (\theta - \theta^*), \tag{32}$$

and a step size update with quantization noise $\varepsilon$ whose mean is $\mu_\varepsilon = \mathbb{E}[\varepsilon] \neq 0$:

$$\theta_{t+1} = \theta_t - \eta \big( \nabla L(\theta_t) + \varepsilon \big) \implies \mathbb{E}[\theta_{t+1}] = \mathbb{E}[\theta_t] - \eta \big( \lambda (\mathbb{E}[\theta_t] - \theta^*) + \mu_\varepsilon \big). \tag{33}$$

Define the error

$$e_t \triangleq \mathbb{E}[\theta_t] - \theta^*. \tag{34}$$

Then

$$e_{t+1} = \mathbb{E}[\theta_{t+1}] - \theta^* = \big[ \mathbb{E}[\theta_t] - \eta (\lambda e_t + \mu_\varepsilon) \big] - \theta^* = e_t - \eta \lambda e_t - \eta \mu_\varepsilon. \tag{35}$$

Therefore,

$$e_{t+1} = (1 - \eta \lambda) e_t - \eta \mu_\varepsilon. \tag{36}$$

Unrolling this recursion gives, for $a = 1 - \eta \lambda$,

$$e_n = a^n e_0 - \eta \mu_\varepsilon \sum_{k=0}^{n-1} a^k. \tag{37}$$

Since $\sum_{k=0}^{n-1} a^k = \frac{1-a^n}{1-a}$ and $1 - a = \eta \lambda$, we get

$$e_n = a^n e_0 - \frac{\eta \mu_\varepsilon}{\eta \lambda} \big( 1 - a^n \big) = a^n e_0 - \frac{\mu_\varepsilon}{\lambda} \big( 1 - a^n \big). \tag{38}$$

The loss at step n is

$$L_n = L(\mathbb{E}[\theta_n]) = \tfrac{1}{2} \lambda e_n^2 = \frac{\lambda}{2} \Big( a^n e_0 - \frac{\mu_\varepsilon}{\lambda} (1 - a^n) \Big)^2. \tag{39}$$

As $n \to \infty$, $a^n \to 0$ (for $a < 0$, which is required for successful optimization), yielding the stationary error and residual loss

$$e_\infty = -\frac{\mu_\varepsilon}{\lambda}, \qquad L_\infty = \frac{\mu_\varepsilon^2}{2\lambda}. \tag{40}$$

Thus, instead of converging to $\theta^*$ with zero loss, biased SGD settles at

$$\mathbb{E}[\theta_\infty] = \theta^* - \frac{\mu_\varepsilon}{\lambda}, \tag{41}$$

and leaves a residual loss

$$L(\mathbb{E}[\theta_\infty]) = \frac{\mu_\varepsilon^2}{2\lambda}. \tag{42}$$

# E  ALGORITHMS

## E.1  AUXILARY ALGORITHMS

In this subsection, we present the algorithms for computing the PLUMAGE probabilities (Algorithm 2) and sampling from it exactly $k$ indices (Algorithm 3) and constructing the projection matrices (Algorithm 4) as discussed in Sections 3 and 3.2. Note that all algorithms can be computed in $(O(\min(m, n)))$. This includes the reverse lookup via sorted search in Algorithm 3 that in case of naive implementation, requires $O(r\log(\min(m, n)))$; however since both arm pointers and target array are sorted, the implementation can continue to iterate over the target array as it locates the arm positions in sub-linear time.

---

**Algorithm 2** COMPUTE_SAMPLING_PROBABILITIES$(\boldsymbol{\sigma}, k, \varepsilon)$

---

1: **Inputs:** $\boldsymbol{\sigma} \in \mathbb{R}^n$ (singular values, descending order), target rank $k$, numerical tolerance $\varepsilon$
2: **Output:** deterministic rank $r^*$, inclusion probabilities $\boldsymbol{p} \in \mathbb{R}^d$
3: $\mathbf{t} \leftarrow \mathbf{reverseCumSum}(\boldsymbol{\sigma})$      $\triangleright\, t_i = \sum_{j=i}^d \sigma_j$
4: $\mathbf{t} \leftarrow \max(\mathbf{t}, \varepsilon)$      $\triangleright$ clip to avoid division by 0
5: $\mathbf{i} \leftarrow (0, 1, \ldots, n-1)$      $\triangleright$ rank indices
6: $\mathbf{s} \leftarrow k - \mathbf{i}$      $\triangleright$ scaling factors $(k - r_t)$
7: $\mathbf{q} \leftarrow (\mathbf{s} \odot \boldsymbol{\sigma}) \oslash \mathbf{t}$      $\triangleright$ test scores $q_i = \frac{(k-i)\sigma_i}{\sum_{j>i}\sigma_j}$
8: $c \leftarrow |\{i \mid q_i < 1\}|$      $\triangleright$ count how many ranks pass the test
9: $r^* \leftarrow n - c$      $\triangleright$ minimal rank that always enters $(r^* = d - c)$
10: $\boldsymbol{p}_{0:r^*-1} \leftarrow 1$      $\triangleright$ deterministic inclusion for top $r^*$ modes
11: $\boldsymbol{p}_{r^*:n-1} \leftarrow \dfrac{(k-r^*)\,\boldsymbol{\sigma}_{r^*:n-1}}{t_{r^*}}$      $\triangleright$ stochastic inclusion for remaining modes
12: **return:** $r^*, \boldsymbol{p}$

---

**Algorithm 3** "Wheel of Fortune": SAMPLE_INDICES$(\boldsymbol{p}, k)$

---

1: **Inputs:** inclusion probabilities $\boldsymbol{p} \in \mathbb{R}^n$, target sample size $k$
2: **Output:** index set $I \subseteq \{0, \ldots, n-1\}$, $|\mathcal{I}| = k$
3: $\boldsymbol{\tau} \leftarrow \mathbf{randPerm}(n)$      $\triangleright$ shuffle the indices
4: $\boldsymbol{p}^{\mathrm{sh}} \leftarrow \boldsymbol{p}[\boldsymbol{\tau}]$      $\triangleright$ permute the probabilities
5: $\mathbf{c} \leftarrow \mathbf{cumSum}(\boldsymbol{p}^{\mathrm{sh}})$      $\triangleright\, c_i = \sum_{j=0}^i p_j^{\mathrm{sh}}$
6: $\Delta \leftarrow \frac{\sum_{i=1}^n p_i^{\mathrm{sh}}}{k}$      $\triangleright$ step size = total mass divided by $k$
7: $\beta \leftarrow \mathbf{Uniform}(0, \Delta)$      $\triangleright$ random offset
8: $\mathbf{u} \leftarrow (0, \Delta, \ldots, (k-1)\Delta) + \beta$      $\triangleright$ $k$ equally spaced pointers
9: $\mathbf{j} \leftarrow \mathbf{searchSorted}(\mathbf{c}, \mathbf{u})$      $\triangleright$ first index with $c_j \geq \mathbf{u}$
10: $I \leftarrow \boldsymbol{\tau}[\mathbf{j}]$      $\triangleright$ map back to original indices
11: **return** $I$

---

**Algorithm 4** SAMPLE_PROJECTIONS$(\mathbf{U}, \boldsymbol{p}, r)$

---

1: **Inputs:** $\mathbf{U} \in \mathbb{R}^{m \times n}$ all singular vectors, column-stacked matrix (assuming $m \leq n$), $\boldsymbol{p} \in \mathbb{R}^n$ sampling probabilities, target rank $r$
2: **Output:** projections $\mathbf{P} \in \mathbb{R}^{m \times r}$ and $\mathbf{D} \in \mathbb{R}^{r \times r}$
3: $I \leftarrow$ sample_indicies$(\boldsymbol{p}, r)$
4: $\mathbf{P} \leftarrow \mathbf{U}[:, I]$
5: $\mathbf{D} \leftarrow \mathrm{Diag}(\boldsymbol{p}[I])$
6: **return** $\mathbf{P}, \mathbf{D}$

---

Table 8: Wall run-time comparison for PLUMAGE, GALORE and ADAM in GPU hours

|              | 350M | 1B    |
|--------------|------|-------|
| AdamW        | 53.9 | 466.7 |
| Galore       | 65.8 | 684.3 |
| Plumage S/MP | 65.3 | 687.2 |

Table 9: rational improvement: optimization gap compared to full-rank optmization and number of steps to surpass the terminal evaluation loss of Galore.

| Method           | PT-130M  | PT-350M  | PT-1B     | FT*-8B  |
|------------------|----------|----------|-----------|---------|
| ADAM             | 25.95    | 19.02    | 14.30     | 2.16    |
| Galore           | 30.18    | 24.08    | 17.03     | 2.86    |
| Plumage S/MP     | 28.73    | 21.81    | 16.29     | 2.64    |
| Improvement rate | 34.28%   | 44.86%   | 27.11%    | 31.43%  |
| Steps            | 15K/20K  | 30K/60K  | 70K/100K  | 2K/13K  |

### E.2 PLUMAGE ALGORITHM AND COMPUTATIONAL COMPLEXITY

The main plumage algorithm is given in Algorithm 5. The computational cost is dominated by SVD ($O(d^3)$ assuming gradients are $\mathbb{R}^{d \times d}$ similarly to GALORE), which is hard to accelerate in hardware compared to matrix-matrix multiply (GEMM) since it requires sequential matrix-vector products. As mentioned in the previous section, the complexity of computing $p$ and sampling from it (Algorithm 4, Algorithm 2) is $O(d)$. We perform both algorithms after each SVD. Thus, the baseline PLUMAGE computational overhead is marginal compared to that of GALORE. In addition, State update methods ($S/MP$, Eqs. (18) and (19)) are dominated by GEMM operations with complexity $\mathbf{M}^\perp : O(d^2 r)$, $\mathbf{B}$ and $\mathbf{V}^\perp : O(d^2 r)$. These operations are easy to accelerate on modern GPUs and are done once per SVD, amortizing their cost. Furthermore, in our experiments using A100/A6000 GPUs, we found that the total training time of our method, using unoptimized implementations, is unaffected by sampling and aligning of moments due to the dominance of SVD overhead. The memory footprint is similar to GaLoRE. It differs by the additional $r$ sampling scale factors per layer. These can be offloaded to the host and prefetched before optimizer weight updates to maintain GaLoRE's memory footprint. Finally, the adaptive SVD interval method ($S/MP/A$) requires computing the correlation matrix $\mathbf{P}_2^\top \mathbf{P}_1 \in \mathbb{R}^{r^2} : O(r^2 d)$ and SVD over the smaller matrix $O(r^3)$. This is done every time the SVD is computed on a new gradient, and the additional SVD step on the small $r \times r$ matrix can be done asynchronously to leverage underutilized compute time since it does not impact the weight update without concurrently storing the old and the fresh projection matrices.

We observed no meaningful wall-clock time difference between the methods, which is expected since we simply replaced the estimation method without changing the optimizer. Our sampling and realignment strategies are much lighter than the SVD operation and are amortized over the SVD interval, leading to negligible overhead. We extended. Moreover, the optimization improvements naturally imply that our method can reach Galore terminal loss much earlier (and without tuning the full-rank optimizer hyperparameters), as can be seen in Fig. 8.

In Table 8, we present a comparison table showing the unoptimized wall runtime on dgx-A100-40GB in total GPU hours (torch compile enabled, using cuda pytorch docker 24:08 and 8-way distributed data parallel training).

In Table 9 we present relational gains in the optimization gap (where the improvement ratio = 1 - $\frac{\text{PPL}_{\text{Plumage}} - \text{PPL}_{\text{Adam}}}{\text{PPL}_{\text{Galore}} - \text{PPL}_{\text{Adam}}}$). Additionally, we compare the efficiency of the training in terms of the number of steps required to reach the terminal loss of the original method (since we showed that the computational cost difference is negligible Table 8). We measure the eval statistics every 5k steps for pretraining and every 2k steps for FT. These results demonstrate a consistent and significant gain across model sizes.

## F EXPERIMENTAL SETTINGS

**Algorithm 5** Adam with PLUMAGE

**inputs:** linear layer weight $\mathbf{W} \in \mathbb{R}^{m \times n}$ with $m \leq n$, scalar loss function: $\mathcal{L} : \mathbb{R}^{m \times n} \to \mathbb{R}^1$, first and second moment decay rates $\beta_1, \beta_2$, learning rate $\eta$, target rank $r$, number of optimization steps $N$, $\tau_0$ number of steps between SVD, $\kappa$ number of steps to resample projection.
**initialize:** $\mathbf{M}_0, \mathbf{V}_0 \in \mathbb{R}^{n \times r}, \mathbf{P}_0 \in \mathbb{R}^{m \times r} \leftarrow I, \leftarrow 0, t \leftarrow 0$.
**repeat**
   $\mathbf{G}_t \leftarrow \nabla_{\mathbf{W}} \mathcal{L}(\mathbf{W}_t)$
   **if** $t \mod \tau_t = 0$ **or** $t \mod \kappa = 0$ **then**
      **if** $t \mod \tau_t = 0$ **then**
         $\mathbf{U}_t, \boldsymbol{\sigma}_t, \mathbf{V}_t \leftarrow \text{SVD}(\mathbf{G}_t)$
         $r_t^*, \boldsymbol{p}_t \leftarrow \text{compute\_sampling\_probabilities}(\boldsymbol{\sigma}_t, r, \varepsilon = 1e - 12)$
      **else**
         $\mathbf{U}_t, \boldsymbol{\sigma}_t, \mathbf{V}_t \leftarrow \mathbf{U}_{t-1}, \boldsymbol{\sigma}_{t-1}, \mathbf{V}_{t-1}$
      **end if**
      $\mathbf{P}_t, \mathbf{D}_t \leftarrow \text{sample\_projection}(\mathbf{U}_t, \boldsymbol{p}_t, r)$            $\triangleright$ Algorithm 4
      $\mathbf{M}_t, \mathbf{V}_t \leftarrow \text{update\_state}(\mathbf{M}_t, \mathbf{V}_t, \mathbf{P}_t, \mathbf{P}_{t-1})$          $\triangleright$ Eqs. (18) and (19)
      $\tau_{t+1} \leftarrow \text{update\_interval}(\tau_t, \mathbf{P}_t, \mathbf{P}_{t-1})$            $\triangleright$ Algorithm 1
   **else**
      $\mathbf{P}_t, \mathbf{D}_t \leftarrow \mathbf{P}_t, \mathbf{D}_{t-1}$
   **end if**
   $\mathbf{R}_t \leftarrow \mathbf{P}^\top \mathbf{G}_t$
   $\mathbf{M}_t \leftarrow \beta_1 \cdot \mathbf{M}_{t-1} + (1 - \beta_1) \cdot \mathbf{R}_t$
   $\mathbf{V}_t \leftarrow \beta_2 \cdot \mathbf{V}_{t-1} + (1 - \beta_2) \cdot \mathbf{R}_t^{\circ 2}$
   $\mathbf{Z}_t \leftarrow \frac{\sqrt{1 - \beta_2^t}}{1 - \beta_1^t} \cdot \frac{\mathbf{M}_t}{\sqrt{\mathbf{V}_t} + \epsilon}$
   $\mathbf{W}_t \leftarrow \mathbf{W}_t - \eta \cdot \mathbf{P}_t \mathbf{D}^{-1} \mathbf{Z}_t$
   $t \leftarrow t + 1$
**until** $t = N$

(a) LLama-130M train loss      (b) LLama-350M train loss      (c) LLama-1B train loss

(d) LLama-130M validation loss    (e) LLama-350M validation loss    (f) LLama-1B validation loss

Figure 8: Pretraining loss plots of Llama on C4 datasts.

Our pre-training experiment models are variants of Llama (Touvron et al., 2023), as suggested by Zhao et al. (2024). These models utilize the same meta-architecture as Llama, except for RMS normalization layers (Zhang & Sennrich, 2019) and SwiGLU activation functions (Shazeer, 2020). The exact model configuration

Table 10: Model configurations.

| Configuration | 130M | 350M | 1B |
|---|---|---|---|
| Depth | 12 | 24 | 24 |
| Rank/Hidden | 256/768 | 256/1024 | 512/2048 |
| Intermidiate | 2048 | 2736 | 5461 |
| Heads | 12 | 16 | 32 |

is given in Table 10. In all experiments, we used a single-node GPU server with 8xA100-40GB GPUs or 8xA100-80GB GPUs. The total compute time per experiment varied from $\sim 9$ GPU hours for the small 130M parameter model pre-training to $\sim 620$ GPU hours for the larger 1B model. We used a slightly modified version of Huggingface transformers Wolf et al. (2019) causal model training and GLUE finetuning code examples. We keep the model weights, gradients, and projections in the FP32 data type while using the BF16 mixed-precision support. In Fig. 8 we present the full validation and pre-training loss plots for pre-training experiments.

## G    IMPLEMENTATION DIFFERENCES WITH GALORE PRETRAINING CODE

**Data pre-processing.** The reference implementation uses padded/truncated sequences that contain about 30% padding tokens per batch from the allenai/c4-en dataset. In contrast, the run_clm.py script from HuggingFace examples (Wolf et al., 2019) uses continuous text chunks of 256 tokens, potentially from different documents for training and evaluation. This can lead to higher perplexity overall due to the challenge of accurately predicting the first tokens of a new sequence with unrelated context.

**Included layers.** Zhao et al. (2024) reports results with the final classification layer training in full-rank, while we state that all linear layers are using low-rank optimization (including the final layer). This is a significant difference resulting in a more challenging setting where the final layer cannot compensate for earlier layers.

**Evaluation loss implementation.** In the reference implementation, loss is reported as the average per batch loss, and not the average loss of the evaluation set. Moreover, we use a larger set of 67M tokens vs. 10M tokens. We find that batch-averaged is constantly lower than the global token average loss.

To show the comparability of our results, we confirmed that the reference code can produce Galore's results on Llama-350M. Then, we trained new models using the reference code with Plumage and Galore estimators, while including lm_head weights, which were omitted from the low-rank parameter set in Galore. The results are presented in Table 11.

Table 11: Training results using Galore's reference code including lm_head weights.

| Estimator | Eval Token Loss | Eval Token PPL |
|---|---|---|
| Full-Rank (Adam) | 2.93 | 18.80 |
| Galore + Adam | 3.099 | 22.18 |
| Plumage + Adam | 2.993 | 19.95 |

Thus, the optimization gap improvement ratio is 65% compared to 44% we presented in Table 4 of the paper. Next, we fix the evaluation loss metric discrepancy in the reference implementation (so metrics accurately represent the full evaluation data). Then, we compare our original checkpoints (from Table 3) over the reference dataset construction (i.e., with padding). These comparisons are shown in Table 12.

Table 12: Comparison of the original checkpoints from Table 4 on Galore's reference dataset (with padding).

| Estimator | Eval Token Loss | Eval Token PPL | Reported PPL (Table 4 ) |
|---|---|---|---|
| Galore + Adam | 3.069 | 21.53 | 24.08 |
| Plumage + Adam | 3.008 | 20.26 | 21.81 |

These findings support the described results and the associated conclusions.

