# OpenReview forum: "PLUMAGE: probablistic low-rank unbiased min variance gradient estimation framework for efficient large model training"
_ICLR.cc/2026/Conference — Submitted to ICLR 2026_

### Official Review · Reviewer_NWtU · 2025-10-20

**Soundness:** 3
**Presentation:** 2
**Contribution:** 2
**Rating:** 6
**Confidence:** 4

**Summary:**

The paper introduces PLUMAGE, a probabilistic low-rank gradient estimator designed to enable efficient large language model (LLM) training by unifying unbiasedness, minimum variance, and adaptive subspace projection.
Unlike prior low-rank optimizers such as GALORE (deterministic top-\(k\) truncation) or FLORA (random Gaussian projection),
 PLUMAGE samples singular components probabilistically with analytically optimal inclusion probabilities, ensuring that the expected reconstructed gradient equals the true gradient while minimizing estimator variance.
The method further incorporates a one-sided projection to reduce memory footprint, a moment realignment scheme to stabilize stateful optimizers when switching subspaces, and an optional adaptive update interval \( \tau \) to balance computation and adaptivity.
Empirical results across language modeling and fine-tuning benchmarks show that PLUMAGE achieves faster convergence and lower perplexity than prior low-rank and stochastic baselines.

**Strengths:**

1. The paper introduces a probabilistic formulation for low-rank gradient estimation that unifies unbiasedness and variance minimization under the MVUE framework. This formulation advances beyond prior deterministic (GALORE) and stochastic (FLORA) approaches by analytically deriving optimal sampling probabilities for singular components and incorporating stabilizing mechanisms such as moment realignment and adaptive projection intervals, effectively balancing bias, variance, and efficiency in subspace-based gradient compression.

2. The paper provides extensive empirical validation across both pretraining and fine-tuning tasks. Its contributions are practically significant: PLUMAGE achieves comparable or superior performance while reducing memory footprint and computational overhead.

**Weaknesses:**

1. The methodological exposition lacks clarity and completeness. The probabilistic sampling process, moment realignment mechanism, and adaptive projection schedule are described conceptually but not operationally, leaving ambiguity in how these components interact in practice. A more detailed algorithmic breakdown or pseudocode would improve reproducibility and interpretability.

2. The bias–variance–adaptivity analysis, though theoretically motivated, is not convincingly supported by empirical data. The paper would benefit from concrete diagnostics or visual evidence—such as gradient–subspace alignment plots, variance trajectories, or bias decomposition—to demonstrate that PLUMAGE achieves the claimed statistical balance during optimization.

3. The empirical results are not sufficiently competitive with current state-of-the-art baselines. The reported improvements over GALORE are modest, and there is no direct comparison with FLORA, which is a closely related stochastic low-rank optimization method. Moreover, the evaluation focuses mainly on perplexity, limiting insight into generalization and robustness. Including comparisons with FLORA under equivalent conditions, as well as additional downstream benchmarks, would make the empirical validation more comprehensive and convincing.


4. The scope of efficiency is narrowly defined. PLUMAGE focuses primarily on computation and memory efficiency but does not consider communication efficiency or distributed scalability, which limits its practical applicability in large-scale multi-GPU or distributed training scenarios.

**Questions:**

Q1: PLUMAGE operates within a fundamental bias–variance–adaptivity trade-off:  Switching subspaces enables exploration of diverse gradient directions and reduces bias but increases variance, while fixing a single subspace lowers variance at the cost of stale, biased gradients that cannot capture evolving curvature. Although the paper claims to balance these effects through probabilistic linear projection and periodic realignment, it remains unclear how this mechanism explicitly mitigates bias without constraining directional diversity. Could the authors clarify whether any quantitative metric (e.g., gradient–subspace alignment, principal-angle drift, or bias–variance decomposition) or visualization (such as alignment trajectories or spectral evolution) is available to demonstrate that PLUMAGE indeed achieves an optimal trade-off between bias, variance, and adaptivity?

Q2: GALORE deterministically selects the top-\(k\) singular components via SVD, preserving gradient fidelity but introducing bias, whereas PLUMAGE stochastically samples \(k\) components using the “Wheel-of-Fortune” mechanism. However, this raises concerns about sampling instability—what if, by chance, low-energy (“bad luck”) components are selected, causing large deviation from the true gradient? The paper should clarify how such cases are prevented or bounded and provide justification for why this probabilistic selection can outperform deterministic top-\(k\) truncation. Is the improvement primarily due to reduced bias, and is this the main reason PLUMAGE requires fewer training steps compared with GALORE? A quantitative or theoretical comparison between probabilistic and monotonic top-\(k\) selection would substantially strengthen this claim.

Q3: PLUMAGE samples gradient components probabilistically, enforcing unbiasedness by matching the expected subspace to the full gradient and minimizing estimator variance. However, the FLORA paper also performs stochastic decomposition, treating LoRA as a gradient compressor through random Gaussian projections. What, then, is the fundamental distinction between PLUMAGE’s probabilistic sampling and FLORA’s stochastic approach, is it primarily the variance minimization objective? Furthermore, given that PLUMAGE only samples a fixed budget of components, is the full SVD decomposition strictly necessary? In principle, FLORA’s random-projection method might offer comparable computational efficiency with lower overhead. The paper would benefit from a direct quantitative comparison with FLORA under the same stochastic-variance setting to justify the need for SVD-based probabilistic sampling and its claimed advantages in variance reduction and convergence stability.


Q4: PLUMAGE derives an unbiased, minimum-variance gradient estimator by taking an expectation over the stochastic sampling of singular components. However, in practice, this expectation is never explicitly computed—the unbiasedness is instead guaranteed through the projection mechanism, where each sampled component is reweighted by its inclusion probability \(p_i\). Could the authors clarify how this projection-based design ensures the expectation property is maintained throughout training, especially when projections are reused over multiple steps? Moreover, does this expectation formulation translate into tangible variance reduction during optimization, or is the benefit primarily theoretical? Empirical evidence—such as variance trajectories or gradient–projection alignment metrics—would help validate that the projection mechanism indeed achieves the claimed minimum-variance behavior in practice.



Q5: PLUMAGE emphasizes using only a left-sided projection matrix for gradient estimation, unlike prior low-rank compression methods such as  PowerSGD[1] that employ both left and right projections. Could the authors clarify whether a symmetric right-sided counterpart exists—i.e., would applying a right-sided projection yield an equivalent estimator in expectation? Furthermore, what is the theoretical or empirical rationale for retaining only the left projection? Does this one-sided formulation fully preserve the statistical properties (e.g., unbiasedness and minimum variance) of the two-sided estimator, or might it introduce asymmetry or a loss in representational fidelity? Providing a short theoretical justification or ablation demonstrating the equivalence between left- and right-sided projections would strengthen the validity of this design choice and its practical implications.



Q6: The paper claims that PLUMAGE can employ an adaptive update interval \( \tau \) to dynamically control how often the SVD and projection subspace are refreshed, yet no quantitative or ablation comparison with a fixed \( \tau \) is presented. Could the authors clarify how this adaptive mechanism is implemented in practice—specifically, what metrics or thresholds determine when the projection is updated—and whether it leads to measurable efficiency or convergence benefits compared to a fixed interval? In the absence of empirical evidence, it remains unclear whether the adaptive \( \tau \) contributes to meaningful computational savings or improved training stability. A direct comparison between adaptive and fixed \( \tau \) settings would help validate this claim.


Q7: Although PLUMAGE is presented as an efficient training approach for large language models, the paper primarily reports validation perplexity as the key metric for evaluating optimization quality and performance. However, perplexity alone may not fully capture broader aspects such as generalization, calibration, or downstream task performance, even if this evaluation setting is consistent with GALORE. Could the authors justify why perplexity is considered a sufficient proxy for optimization progress in this work, and whether improvements in perplexity reliably translate to better model capability? Including complementary evaluations—such as downstream task accuracy, calibration error, or convergence efficiency—would help substantiate the claim that PLUMAGE achieves meaningful training efficiency beyond loss-level improvements.

Q8: While PLUMAGE is presented as an efficient optimization framework, the paper appears to focus primarily on reducing computational and memory costs (e.g., via low-rank projection, one-sided estimation, and adaptive SVD updates) rather than communication overhead in distributed training. Could the authors clarify whether PLUMAGE is intended solely as a computation-efficient method, or if it can also be extended to communication-efficient scenarios such as multi-GPU or distributed optimization? Additionally, have the authors considered evaluating the method’s impact on communication volume or synchronization cost to substantiate its scalability claims in distributed environments?

[1]Vogels, Thijs, Sai Praneeth Karimireddy, and Martin Jaggi. "PowerSGD: Practical low-rank gradient compression for distributed optimization." Advances in Neural Information Processing Systems 32 (2019).

This paper deserves a certain degree of discussion. Although the current version is not yet sufficient for acceptance, it presents several promising ideas that could develop into a strong contribution with further clarification and empirical support. I have raised multiple questions (Q1–Q8) regarding the methodology, empirical validation, and evaluation scope, and there remain additional issues beyond those listed. Nevertheless, I am willing to give the authors the benefit of the doubt at this stage, as the proposed framework shows potential and relevance for advancing efficient large-scale model optimization.

---

> ### Author Response · Authors · 2025-12-03
> **NWtU**
>
> * Please note that all algorithms are available in Appendix E. This includes the probabilistic sampling procedure and the computation of the sampling probabilities (Algorithms 2-4). Additionally, momentum alignment formulas (Eq. 18 & 19) are defined mathematically. The full Plumage+ADAM is given in Algorithm 5 (see pointer at the end of section 3).
>
>   1. As we explain in Section 3, when the subspace is periodically updated, the variance optimality of this solution is not guaranteed. However, the consistency of the gradient subspace (e.g., Figure 5 in the Appendix, where the mean principal angles between subsequent projections are in the range of 40-60%, while Figure 4 suggests the overlap ratio is consistent in each layer during training) supports the plausible assumption that PLUMAGE offers variance reduction benefit under periodic sampling.
>       * For completeness of the ablation, we added a small experiment to show that Plumage MVUE sampling outperforms a naive unbiased random sampling while visibly reducing variance over 5 seeds.
>
>   2. We do not consider low-probability directions to be "bad" and make no attempt to stop Plumage from using them.
>   That said, one can easily adjust the sampling probability by truncating probabilities (i.e., excluding such directions at the cost of accepting small bias) or setting a floor value (e.g., SVD error bounds, addressing numerical accuracy of small SV). See our previous responses regarding the importance of the tail spectra.
>
>   3. Unlike Plumage, Flora samples a fully random subspace. This means that the added variance of the estimator is proportional to the size of the projection matrices. While both methods are unbiased, there is no evidence that Flora can be competitive for training large models (See Table 2 in [1], pretraining with random projection is worse than Galore). Furthermore, we do not change the training regime other than the gradient estimation method. We report the best results on GLUE following a tracktable grid search, based on the author's hyperparameters. However, there are no published hyperparameters or implementation of Flora+ADAM for pretraining. Our best efforts to create a comparable setup were unsuccessful (as supported by results from [1]).
>
>   4. We do not explicitly compute the expectation of Plumage during low-rank training. However, it can be implemented for distributed training as described in the discussion section. We show in Section 4.2 that when combined with an unbised optimizer (SGD+Momentum), the training loss of Plumage+SGD+M tracks the loss of the full-rank optimizer, while the loss of Galore+SGD+M diverges even with a fixed update interval of 200 steps. We hypothesize that the errors with periodic sampling are less likely to be correlated and accumulate over time compared to using the top-k spectra.
>
>   5. Plumage can use either left or right side projections. The point we made was in relation to previous work, such as ATOMO [4], where we only need a one projection matrix (left or right) for each linear layer's gradients (Note that POWERSGD [2] is limited to the top spectra as it uses power iteration to avoid a full SVD).
>
>   6. In Appendix C, we discuss a method to evaluate the fitness between the new gradient and the current projection and how one can leverage it to either improve subspace tracking or potentially accelerate training by automatically determining when SVD should be computed. This expands on the adaptive interval method discussed in previous work that relied only on the leading component correlation. This approach requires HP tuning for each objective and training scenario; thus, we left it outside the scope of our current work and include it as part of our auxiliary discussion and results.
>
>   7. We use the common evaluation practices in the related literature. We report a wide range of metrics on various widely accepted benchmarks such as GLUE and commonsense (not just loss/perplexity). Additionally, the loss is a key metric since we are interested in measuring the impact of gradient estimation on the **optimization** process.
>
>   8. Plumage is a general Low-rank gradient estimation method with various practical applications. We focus on well-established training regimes and demonstrate that in the challenging task of optimization with strictly low-rank gradients, we can improve the optimization by simply replacing the estimator without changing the regime or tuning additional hyperparameters. However, Plumage can also offer gains in communication reduction (since it is the only fixed-size min-variance unbiased low-rank estimator that can recover full-rank gradient).
>
>   [1] NAACL 2025 - CompAct: Compressed Activations for Memory-Efficient LLM Training
>
>   [2] NeurIPS 2019 - PowerSGD: Practical low-rank gradient compression for distributed optimization
>
>   [3] ICLR 2025 - DOES SGD REALLY HAPPEN IN TINY SUBSPACES?
>
>   [4] Neurips 2018 - Atomo: Communication-efficient Learning via Atomic Sparsification

---

### Official Review · Reviewer_qx4C · 2025-10-31

**Soundness:** 3
**Presentation:** 2
**Contribution:** 2
**Rating:** 4
**Confidence:** 5

**Summary:**

This paper introduces PLUMAGE, a novel framework for memory-efficient training of large models that addresses key limitations in existing low-rank gradient methods. The authors identify two primary issues: 1) the bias introduced by deterministic top-k gradient projections, and 2) the optimizer state misalignment. To solve this, PLUMAGE proposes a probabilistic, unbiased, and minimum-variance gradient estimator based on an efficient fixed-rank sampling strategy. Furthermore, it introduces a statistics realignment method to correctly transform the first and second moments of stateful optimizers into the new subspace. Experiments show that PLUMAGE closes the performance gap to full-rank ADAM compared with other low-rank-based methods.

**Strengths:**

1. The paper theoretically proposes an unbiased minimum-variance low-rank gradient estimator (PLUMAGE) and applies it to the efficient training of large-scale language models. This approach is theoretically inspiring and provides a new perspective on solving the bias problem in low-rank gradient estimation.
2. The paper introduces several practical and insightful contributions. The "wheel-of-fortune" sampling algorithm is an interesting and efficient trick for the k-sparse sampling problem. More importantly, the paper correctly identifies that optimizer state realignment is a critical and often overlooked issue in low-rank optimization methods that use periodic projection updates. The proposed realignment strategy (Eq. 18 and 19) provides a well-reasoned potential solution to this problem, which is highly inspiring for future work in this area.

**Weaknesses:**

1. Minor Typos in Appendix: The analysis in Appendix A, while ultimately arriving at a standard and correct result (Eq. 22), contains several confusing typos. Specifically, Equation (21) misrepresents the matrix inner product within the Trace. Maybe it should be $v_i u_i^\top u_j v_j^\top$ in the left one and $v_i v_i^\top$ in the right one.
2. The paper's primary contribution is its formulation as an unbiased minimum-variance estimator, distinguishing it from biased top-k methods like GALORE. However, the practical significance of this distinction is worth discussing given the known properties of LLM gradients. The premise of methods like GALORE is that LLM gradient spectra are "top-heavy," with rapid decay, meaning the top-k singular vectors capture the vast majority of the gradient's energy and information. This empirical phenomenon is well-documented [https://arxiv.org/abs/1611.07476], [https://arxiv.org/abs/2403.03507]. Under this top-heavy assumption, the PLUMAGE sampling algorithm will likely behave in a predictable way: the deterministic rank will automatically capture these dominant directions by assigning them $p_i=1$ (because for these directions their dominant singular values satisfy $(k-r)\sigma_r \ge \sum_{i=r+1}^n \sigma_i$. Consequently, the sampling process will be truncated: the top $r^{\*}$ components are selected with high probability (identical to top-k), and the remaining $k-r^\*$ rank budget is used to probabilistically sample from the vast "tail" of non-dominant directions. Because the singular values in this tail are small and relatively uniform, the sampling probabilities $p_i$ for these tail components will be small and near-uniform ($p_i \approx (k-r^\*) / (n-r^\*)$).
3. In practice, PLUMAGE's estimator may be functionally equivalent to a standard top-k projection, augmented with adaptive, near-uniform sampling in the non-dominant subspace. While this achieves statistical unbiasedness, the paper does not sufficiently argue why this sampling of the low-energy tail is so impactful, versus simply being a minor algorithmic difference from top-k projection. Therefore, it may be necessary to provide some metrics during the training process and conduct corresponding ablation studies. For example, what is the specific distribution of $p_i$ for different parameters (like attention or MLP parameters) in the early and late stages of training, and does it indeed align with the characteristics of top-k truncation in practice? Alternatively, what are the spectral properties of the full-space gradient in the early and late stages of training? This could affect the method's difference from methods like GALORE, as this difference might emerge during the training process. For instance, does PLUMAGE's advantage stem from exploring more directions in the later stages of training? I believe these points need to be clarified in the paper.
4. The experimental results presented do not show that the PLUMAGE method has a significant advantage. The training curves in Figure 7 suggest that the advantage over GALORE is not significant, and this effect diminishes as the model size increases. Therefore, I am curious about the true effectiveness of the method in larger-scale practical pre-training tasks. Furthermore, PLUMAGE still relies on SVD in Algorithm 5, which is not eliminated. Therefore, when considering the model sharding and parallel strategies required for large-scale experiments, an additional all-gather operation might be necessary to perform SVD on the full gradient, potentially introducing extra communication overhead.

**Questions:**

See Weaknesses.

---

> ### Author Response · Authors · 2025-12-03
> **Response to qx4C**
>
> We update the appendix derivation with no change to the final standard result.
> * **Gradient Spectra discussion** The provided example is wrong, if the spectra is in rapid-decay the distribution of the tail singular values is unlikely uniform (e.g., Marchenko–Pastur SV distribution for a random matrix with $m/n=1$ and small $x$, $f(x)\propto \frac{1}{\sqrt{x}}$). Moreover, Plumage sampling probability will decay at a similar rate (see Equation 11: $p_i
> \propto \sigma_i$). Recall that Plumage sampling is intended to provably minimize variance with respect to a given rank budget regardless of the spectral structure.
> Furthermore, there is evidence of the bulk space for learning [1,2,3,4,5]. The picture suggested by these works (especially in [5]) is that training follows a gently sloped "valley", where the top components are mostly fluctuations in the (less important) directions of the valley, steep directions, while learning requires progress along the (more important) directions of the valley's gentle slope. Moreover, Galore/top-k gradient estimators alone cannot, evidently, recover the training loss of full-rank gradients.
>
> * **Additional metrics** The Appendix contains analysis (see Figures 4,6) based on observing the behaviour of $r^* $. If the spectral composition of the gradient is top-heavy (i.e., given k, $\exists j \leq k$ such that, $\forall i > j:  \sigma_i \ll \sigma_j $), then the optimal min variance solution is $r^* = k$.  In contrast, if rank k is insufficient (to cover the top-spectra) $r^* $ will be smaller to allow sampling to compensate for the lost energy. While this is not the full picture, it serves the purpose of capturing the difference of Plumage from the fixed top-k low-rank estimator.
>   - Figure 4: We show how $r^* $ develops during training for all layers of the 1B LLaMa model. We observe that during the early stage of training, $r^* $ reaches a steady state for all layers. In addition, $r^* $ tends to be much smaller than the fixed rank budget (recall that k=512 out of 2048, we see $r^* <150 $, suggesting top-k pick is far from ideal).
>   - Figure 6: We also calculate the mean (over time) of $r^* $ for each layer, suggesting certain types of layers are more amenable to gradient truncation.
> * **Source of Plumage's advantage:**
>   PLUMAGE is reducing bias by sacrificing a minimal amount of variance. It was shown that bias is typically more harmful to optimization [6,7,8] as we analytically illustrate in Appendix D. Moreover, sampling permits learning in the case of the river valley loss landscape [5]. Furthermore, to the best of our knowledge, Plumage is the only unbiased min-variance fixed-rank estimator that can recover the full-rank gradient (e.g., in distributed settings).
>
> * **Significance of gains:**
>   Recall that Plumage is a general low-rank gradient estimator, whose benefits are shown over alternative low-rank estimators as a plugin replacement -- without changing the established training regime or additional HP tuning.
>   We measure gains as the ratio of improvement in the optimization gap (where Improvement ratio =  $1 - \frac{\mathrm{PPL}\_{\text{Plumage}} - \mathrm{PPL}\_{\text{Adam}}}{\mathrm{PPL}\_{\text{Galore}} - \mathrm{PPL}\_{\text{Adam}}}$). **Alternatively**, training efficiency can be measured in terms of improvement in time to terminal loss of the original method.
>    We have added the following table to clarify this conclusion. Note that we observe a consistent and significant gain across model sizes.
>     | Method           | PT-130M | PT-350M |  PT-1B |  FT*-8B |
>     | ---------------- | :-----: | :-----: | :----: | :----: |
>     | ADAM             |  25.95  |  19.02  |  14.30 |  2.16  |
>     | GALORE           |  30.18  |  24.08  |  17.03 |  2.86  |
>     | PLUMAGES/MP      |  28.73  |  21.81  |  16.29 |  2.64  |
>     | Improvement ratio|  34.28% |  44.86% | 27.11% | 31.43% |
>     | steps  |  15K/20K  |  30K/60K   |  70K/100K      |  2K/13K   |
>     * We measure eval statistics every 5k steps for pretraining and every 2k steps for FT (while fewer steps lessen the potential accumulation of bias)
>     * TL steps are used instead of time since we showed the computational cost difference is negligible in TTT (see table 8, $\ll 1$%).
>
>   Additionally, other factors can impact the optimization gap. For instance, it is reasonable that using a lower rank will increase the gap (since we have to drop more "heavy" spectra components). Alas, we cannot accommodate pretraining of large models with every factor due to the associated costs. We did, however, provide an ablation study (Appendix C).
>
> * **Reliance on SVD:**
>   All components are needed for MVUE sampling weights. However. SVD cost can be mitigated without significant changes to the estimation framework. For example, SVD workload can be offloaded to the CPU or distributed across workers. In Tensor-Parallel, we can treat partitions as independent layers with local solutions (i.e., there is no need for additional allreduce).

---

> > ### Author Response · Authors · 2025-12-03
> > **reference**
> >
> > [1] ICLR 2025 - DOES SGD REALLY HAPPEN IN TINY SUBSPACES?
> >
> > [2] Neurips 2015 - Preconditioned Spectral Descent for Deep Learning
> >
> > [3] Technical report 2025 - Muon is Scalable for LLM Training
> >
> > [4] ICLR 2019 - Gradient descent happens in a tiny subspace
> >
> > [5] ICLR 2025 - Understanding Warmup-Stable-Decay Learning Rates: A River Valley Loss Landscape Perspective
> >
> > [6] CoRR 2015 - Deep learning with limited numerical precision
> >
> > [7] ICLR 2021 - Neural gradients are near-lognormal: improved quantized and sparse training,
> >
> > [8] Neurips 2025 - Fp4 all the way: Fully quantized training of llms

---

### Official Review · Reviewer_ZJNN · 2025-11-01

**Soundness:** 3
**Presentation:** 2
**Contribution:** 3
**Rating:** 4
**Confidence:** 3

**Summary:**

The paper proposes PLUMAGE, a probabilistic low-rank gradient estimator that aims to overcome the bias–variance limitations in existing low-rank optimizers such as GaLore and FLORA. PLUMAGE introduces (1) a k-sparse probabilistic estimator with unbiased minimum-variance sampling and (2) a moment realignment strategy to stabilize training under dynamic projection subspaces. It is designed as a drop-in replacement for existing low-rank optimizers without extra hyperparameters. Experiments across pretraining (LLaMA 130M–1B) and fine-tuning (GLUE and common-sense tasks) show consistent improvements over GaLore.

**Strengths:**

- The experiments design is comprehensive, convering pre-training, fine-tuning (GLUE and common-sense) and multiple ablation studies, providing a complete understanding of the proposed method.

- The framework can be integrated easily with existing optimizers and training pipelines.

- The explanation of the method is clear and easy to understand.

**Weaknesses:**

- The results in Table 4 is strange , as GaLore’s performance is significantly lower than that of the baseline Adam. Based on my experience and community reports, GaLore’s results are reproducible. It would be helpful to analyze this discrepancy and provide detailed hyperparameter settings to clarify the mismatch.

- It would be beneficial to demonstrate end-to-end training efficiency improvements to more convincingly support the efficiency claim.

- Since the proposed method is orthogonal to memory-efficient optimizer designs, it would be interesting to explore their compatibility, i.e., whether there won't be further performance loss. Such results could demonstrate the method’s generality and strengthen its overall impact.

- There are several typos throughout the paper (e.g., lines 102 and 265); careful proofreading is recommended.

**Questions:**

Please see the weakness part.

---

> ### Author Response · Authors · 2025-12-03
> **Response to ZJNN**
>
> * **E2E training efficiency:** We demonstrate Wall-Time equivalence in Appendix Table 8 and discuss the computational complexity in Appendix E.2. Nevertheless, the main result is a low-rank gradient estimation framework for optimization -- without any changes to the established training regime other than the introduction of Plumage instead of the traditional top-k approximator, which is known to minimize MSE.
>   We measure gains as the ratio of improvement in the optimization gap (where Improvement ratio $1 - \frac{\mathrm{PPL}\_{\text{Plumage}} - \mathrm{PPL}\_{\text{Adam}}}{\mathrm{PPL}\_{\text{Galore}} - \mathrm{PPL}\_{\text{Adam}}}$
> . **Alternatively**, training efficiency can be measured in terms of improvement in time to terminal loss of the original method.
> We have added the following table to clarify this conclusion. Note that we observe a consistent and significant gain across model sizes.
>     | Method           | PT-130M | PT-350M |  PT-1B |  FT*-8B |
>     | ---------------- | :-----: | :-----: | :----: | :----: |
>     | ADAM             |  25.95  |  19.02  |  14.30 |  2.16  |
>     | GALORE           |  30.18  |  24.08  |  17.03 |  2.86  |
>     | PLUMAGES/MP      |  28.73  |  21.81  |  16.29 |  2.64  |
>     | Improvement ratio|  34.28% |  44.86% | 27.11% | 31.43% |
>     | steps  |  15K/20K  |  30K/60K   |  70K/100K      |  2K/13K   |
>     * We measure eval statistics every 5k steps for pretraining and every 2k steps for FT (while fewer steps lessen the potential accumulation of bias)
>     * TL steps are used instead of time since we showed the computational cost difference is negligible in TTT ($\ll 1$%).
>
> * **Strange results/GaLore reproduction:** We specifically list implementation differences in Appendix G, while Appendix F covers settings and HP. Below, we discuss the key differences and demonstrate that our results are comparable to or better than those reported when using the reference implementation that reproduces Galore.
>
>   In short, we used Huggingface run_clm.py example code to leverage common practices and accelerate our experiment cycle. The main difference (see Appendix G for more details) is data processing, which involves concatenating examples instead of padding sequences to the max length, which can lead to higher loss. These changes are irrelevant when discussing a general gradient estimator, designed for training LLMs, while Plumage results are presented without any HP tuning to support its superiority to the fixed top-k estimator.
>
>   To show the comparability of our results, we confirmed that the reference code can produce Galore's results on Llama-350M. Then, we trained new models using the reference code with Plumage and Galore estimators, while including lm_head weights, which were omitted from the low-rank parameter set in Galore:
>
>   | Estimator |	eval token loss|	eval token ppl |
>   | :---:     |:------:           | :-----:            |
>   |Full-Rank (Adam) |  2.93 | 18.80 |
>   |Galore+Adam |  3.099 |22.18|
>   |Plumage+Adam | 2.993 |19.95|
>
>   Thus, the optimization gap improvement ratio is 65% compared to 44% we presented in Table 3 of the paper.
>
>   Next, we fix the evaluation loss metric discrepancy in the reference implementation (so metrics accurately represent the full evaluation data). Then, we compare our original checkpoints (from Table 3) over the reference dataset construction (i.e., with padding)
>
>    | Estimator |	eval token loss|	eval token ppl|	reported ppl (table 3) |
>    | :---: |:------:           | :-----:            | :-----:                  |
>   |Galore+Adam | 3.069 | 21.53	 | 24.08 |
>   |Plumage+Adam | 3.008 | 20.26 | 21.81 |
>
>   These findings support the described results in Table 3 and the associated conclusions.
>
> * **Impact outside of memory efficient optimizer design:** Plumage is indeed applicable beyond memory efficient optimizers. We use Plumage+SGD+M to demonstrate that Plumage remains unbiased when using an unbiased optimizer. Other memory-efficient optimizers, such as Apollo [1] or Muon [2], may benefit from extensions of our work that target the bias of the weight update instead of the gradient. Moreover, Plumage's application in distributed settings (e.g., DDP, ZERO3) is likely to benefit optimization over other approach that uses low-rank estimation to reduce communication (since Plumage is the only fixed-size low-rank estimator that can produce a full-rank gradient approximation).
> These settings involve technical challenges and nuances that may warrant a separate manuscript. Thus, we leave this outside the scope of our current work.
>
> [1] MLSys 2025 Apollo: SGD-like memory, AdamW-level performance
>
> [2] Technical report 2025 - Muon is Scalable for LLM Training

---

### Meta-Review · Area_Chair_Q5bv · 2026-01-03

**Summary:**

The reviewers agree that the paper is theoretically sound, but the main disagreement concerns its practical impact for large-scale LLM training. Reviewer qx4C questions whether the proposed probabilistic sampling offers a clear advantage over deterministic top-k methods given the empirically top-heavy gradient spectra of LLMs. In addition, concerns remain about training efficiency and scalability in GPU-centric and distributed settings, where system-level overheads dominate. While the rebuttal clarifies several aspects of the method, it does not fully resolve these two central issues.

**Reviewer Concerns:**

The rebuttal adequately addresses secondary concerns, including implementation details, reproducibility of GaLore results, and minor technical issues in the appendix. These points appear largely resolved.

However, the main concerns raised by Reviewer qx4C remain. First, the rebuttal does not provide sufficiently direct empirical evidence that probabilistic sampling of low-energy gradient components leads to meaningfully different or improved optimization behavior compared to deterministic top-k truncation in realistic LLM training. The argument remains largely theoretical, supported mainly by appendix-level analysis rather than clear training-time diagnostics or large-scale empirical validation.

Second, the efficiency and scalability concerns are not fully addressed. The rebuttal emphasizes step-based efficiency and CPU-side timing, but this does not adequately reflect GPU-dominated, distributed training regimes, where SVD and subspace updates may lie on the critical path and introduce synchronization or communication overhead. The lack of concrete large-scale or distributed experiments makes it difficult to assess whether the method can be applied efficiently in practice.

**Reviewer Scores:**

Given the clarifications in the rebuttal, Reviewer ZJNN might slightly increase their score or remain borderline. Reviewer qx4C’s primary concerns are not sufficiently alleviated, and their score would likely remain unchanged. Reviewer NWtU, while generally positive, also notes unresolved issues regarding empirical validation and scalability and would likely maintain a similar borderline-positive assessment.

---

### Decision · Program_Chairs · 2026-01-26

Reject